# Unifying Feature and Cost Aggregation with Transformers for Semantic and Visual Correspondence

**Sunghwan Hong**[*], **Seokju Cho**[*], **Seungryong Kim**[††]
Korea University
{sung_hwan,seokju_cho,seungryong_kim}@korea.ac.kr

**Stephen Lin**
Microsoft Research Asia
stevelin@microsoft.com

## ABSTRACT

This paper introduces a Transformer-based integrative feature and cost aggregation network designed for dense matching tasks. In the context of dense matching, many works benefit from one of two forms of aggregation: feature aggregation, which pertains to the alignment of similar features, or cost aggregation, a procedure aimed at instilling coherence in the flow estimates across neighboring pixels. In this work, we first show that feature aggregation and cost aggregation exhibit distinct characteristics and reveal the potential for substantial benefits stemming from the judicious use of both aggregation processes. We then introduce a simple yet effective architecture that harnesses self- and cross-attention mechanisms to show that our approach unifies feature aggregation and cost aggregation and effectively harnesses the strengths of both techniques. Within the proposed attention layers, the features and cost volume both complement each other, and the attention layers are interleaved through a coarse-to-fine design to further promote accurate correspondence estimation. Finally at inference, our network produces multi-scale predictions, computes their confidence scores, and selects the most confident flow for final prediction. Our framework is evaluated on standard benchmarks for semantic matching, and also applied to geometric matching, where we show that our approach achieves significant improvements compared to existing methods.

## 1 INTRODUCTION

Finding visual correspondences between images is a central problem in computer vision, with numerous applications including simultaneous localization and mapping (SLAM) (Bailey & Durrant-Whyte, 2006), augmented reality (AR) (Peebles et al., 2021), and structure from motion (SfM) (Schonberger & Frahm, 2016). Given visually or semantically similar images, sparse correspondence approaches (Lowe, 2004) first detect a set of sparse points and extract corresponding descriptors to find matches across them. In contrast, dense correspondence (Philbin et al., 2007) aims at finding matches for all pixels. Dense correspondence approaches typically follow the classical matching pipeline of feature extraction and aggregation, cost aggregation, and flow estimation (Scharstein & Szeliski, 2002; Philbin et al., 2007).

Much recent correspondence research (Sarlin et al., 2020; Sun et al., 2021; Jiang et al., 2021; Xu et al., 2021; Li et al., 2021; Cho et al., 2021; Min et al., 2021a; Huang et al., 2022b; Cho et al., 2022a) have utilized a means to benefit from either feature aggregation or cost aggregation. Feature aggregation, as illustrated in Fig. 1 (a), is a process that aims to not only integrate self-similar features within an image but also align similar features between the two images for matching. The advantages of feature aggregation have been made particularly evident in several attention- and

---

[*]equal contribution
[††]Corresponding author

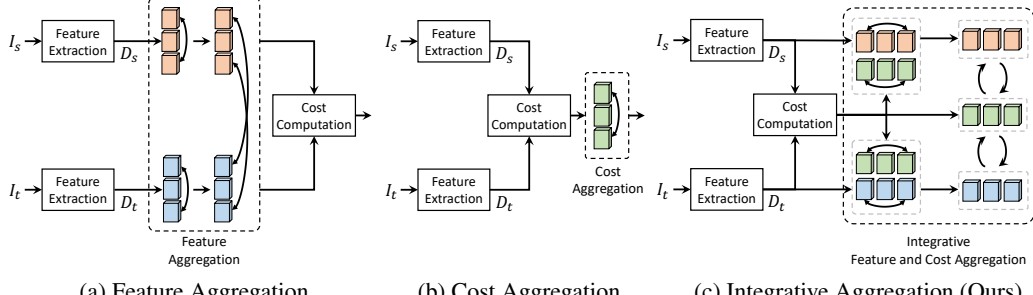

Figure 1: **Intuition of the proposed method:** (a) feature aggregation methods that aggregate feature descriptors, (b) cost aggregation methods that aggregate a cost volume, and (c) our integrative feature and cost aggregation method, which jointly performs both aggregations to find highly accurate correspondences.

Transformer-based matching networks (Vaswani et al., 2017; Sarlin et al., 2020; Sun et al., 2021; Xu et al., 2021; Jiang et al., 2021). Their accuracy in matching can be attributed to, as we show in Fig. 2 (e-f) and supported by previous studies (Sun et al., 2021; Amir et al., 2021), the learned position-dependent semantic features. While the visualization exhibits consistency among parts with the same semantics, dense matching often requires features with even greater discriminative power for more robust pixel-wise correspondence estimation, which is typically challenged by repetitive patterns and background clutters.

To compensate for, on the other hand, cost aggregation, as illustrated in Fig. 1 (b), has been adopted by numerous works (Rocco et al., 2018; Min et al., 2021b; Cho et al., 2021; Huang et al., 2022b; Cho et al., 2022a) for its favorable generalization ability (Song et al., 2021; Liu et al., 2022) and robustness to repetitive patterns and background clutter, which can be attributed to the matching similarities encoded in the cost volumes. These works can leverage the matching similarities to enforce smoothness and coherence in the disparity or flow estimates across neighboring pixels. However, it is important to note that, as highlighted in Fig. 3 (c-h), cost volumes often lack semantic context and exhibit relatively less consideration of spatial structure. This disparity arises due to the fact that the information encapsulated within cost volumes is established on the basis of pixel pairs, which could potentially lead to challenges in scenarios where such contextual cues play a pivotal role.

In this paper, we tackle the dense correspondence task by first performing a thorough exploration of feature aggregation and cost aggregation and their distinct characteristics. We then propose a simple yet effective architecture that can benefit from the potential advantages stemming from a more judicious use of both aggregation processes. The proposed architecture is a Transformer-based aggregation network, namely Unified Feature and Cost Aggregation Transformers (UFC), that models an integrative aggregation of feature descriptors and the cost volume, as illustrated in Fig. 1 (c).

This network consists of two stages, the first of which employs a self-attention layer to aggregate the descriptors and cost volume jointly. In this stage, the descriptors can help to disambiguate the noisy cost volume similarly to cost volume filtering (Hosni et al., 2012; Sun et al., 2018), and the cost volume can encourage the features to account for matching probabilities as an additional factor for alignment. For the subsequent step, we design a cross-attention layer that enables further aggregation aided by the outputs from earlier aggregations. This aggregated cost volume can guide the alignment with its sharpened matching distribution. These attention layers are interleaved. We further propose hierarchical processing to enhance the benefits one aggregation gains from the other. Finally, at inference time, our method estimates multi-scale predictions and their confidence scores to recover highly accurate flow.

We first evaluate the proposed method on the tasks of semantic matching and subsequently, we also substantiate that our framework achieves appreciable performance when applied to geometric matching. Our framework clearly outperforms prior works on all the major dense matching benchmarks, including HPatches (Balntas et al., 2017), ETH3D (Schops et al., 2017), SPair-71k (Min et al., 2019b), PF-PASCAL (Ham et al., 2017) and PF-WILLOW (Ham et al., 2016). We provide extensive ablation and analysis to validate our design choices.

## 2 RELATED WORK

**Feature Extraction and Aggregation.** Feature extraction involves detecting interest points and extracting the descriptors of the corresponding points. In traditional methods (Liu et al., 2010; Bay et al., 2006; Dalal & Triggs, 2005; Tola et al., 2009), the matching performance mostly relies on the quality of the feature detection and description methods, and outlier rejection across matched points is typically determined by RANSAC (Fischler & Bolles, 1981).

Learning-based feature extraction methods (DeTone et al., 2018; Ono et al., 2018; Dusmanu et al., 2019; Revaud et al., 2019) obtain dense deep features tailored for matching. These works have demonstrated that the quality of feature descriptors contributes substantially to matching performance. In accordance with this, recent matching networks (Sarlin et al., 2020; Min et al., 2019a; Lee et al., 2019; Hong & Kim, 2021; Min et al., 2020; Jiang et al., 2021; Sun et al., 2021; Xu et al., 2021) proposed effective means for feature aggregation. Notable sparse correspondence works include SuperGlue (Sarlin et al., 2020) and LOFTR (Sun et al., 2021), which employ graph- or Transformer-based self- and cross-attention for aggregation. Other methods are PUMP (Revaud et al., 2022) and ECO-TR (Tan et al., 2022), which are follow-up works of COTR (Jiang et al., 2021). These methods evaluate on dense correspondence benchmarks in a different way from previous works, using only the sparse and quasi-dense correspondences above certain confidence levels. We refer the readers to the supplementary material for a detailed discussion.

For dense correspondence, SFNet (Lee et al., 2019) and DMP (Hong & Kim, 2021) introduce adaptation layers after feature extraction to learn feature maps well-suited to matching and are evaluated on dense semantic and geometric matching. DKM (Edstedt et al., 2023) adopts Gaussian Processing Kernels for dense correspondence, and it demonstrates its effectiveness in pose estimation. In optical flow, GMFlow (Xu et al., 2021) leverages Transformer for feature aggregation, and its extension Xu et al. (2023) applies the method to stereo matching and depth estimation. In semantic correspondence, notable works include SCorrSAN (Huang et al., 2022a) proposes an efficient spatial context encoder to aggregate spatial context and feature descriptors, and MMNet Zhao et al. (2021) proposes a multi-scale matching network to learn discriminative pixel-level features.

**Cost Aggregation.** In the dense correspondence literature, many works have designed their architectures for effective cost aggregation, which brings strong generalization power (Song et al., 2021; Liu et al., 2022). Recent works (Truong et al., 2020b; Hong & Kim, 2021; Jeon et al., 2020; Truong et al., 2021) use 2D convolutions to establish correspondence while aggregating the cost volume with learnable kernels, while some works (Min et al., 2019a; 2020; Liu et al., 2020) utilize handcrafted methods, which include RHM (Cho et al., 2015) and the OT solver (Sinkhorn, 1967). NC-Net (Rocco et al., 2018) was the first to propose 4D convolutions for cost aggregation, and numerous works (Li et al., 2020; Yang & Ramanan, 2019; Huang et al., 2019; Rocco et al., 2020; Min et al., 2021a;b) leveraged or extended 4D convolutions.

Among Transformer-based networks, CATs (Cho et al., 2021) recently proposed to use Transformer (Vaswani et al., 2017) for cost aggregation, and its extension CATs++ (Cho et al., 2022a) combined convolutions and Transformer for an enhanced cost aggregation. VAT (Hong et al., 2022a) proposed 4D convolutional Swin transformer for cost aggregation that benefits from better generalization power and showed its effectiveness for semantic correspondence. NeMF (Hong et al., 2022b) incorporates an implicit neural representation into semantic correspondence and implicitly represents the cost volume to infer correspondences defined at arbitrary resolution. FlowFormer (Huang et al., 2022b) and STTR (Li et al., 2021) are Transformer-based cost aggregation networks specifically designed for optical flow or stereo matching.

## 3 FEATURE AND COST AGGREGATION

In this section, we examine the characteristics of feature and cost aggregation, which will later be verified empirically in Section 5.3. Fig. 2 and Fig. 3 present visualizations of feature maps and cost volumes at different stages of aggregation. Although there may be different types of aggregations, throughout this work, we focus on attention-based aggregations. From the visualizations, the following observations can be made.

**The information encoded by features and cost volumes differ.** Feature aggregation and cost aggregation thus exploit different information, which is exemplified in Fig. 2 (c-d) and Fig. 3 (c-e)

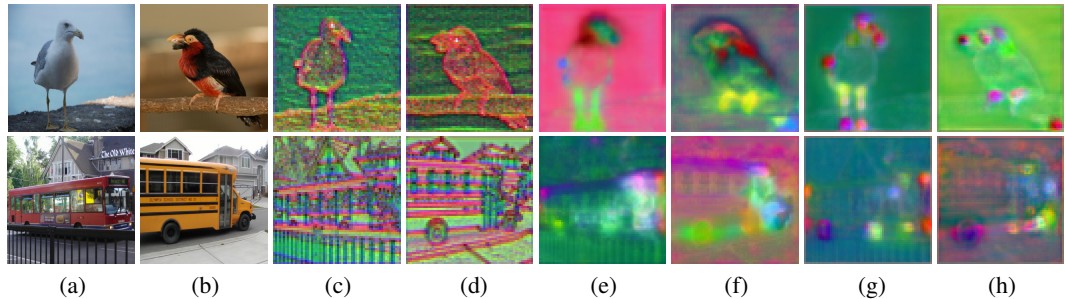

|     |     |     |     |     |     |     |     |
| (a) | (b) | (c) | (d) | (e) | (f) | (g) | (h) |

Figure 2: **PCA visualizations of feature maps:** (a-b) source and target images. (c-d) raw feature maps. (e-f) feature maps that have undergone feature aggregation. (g-h) feature maps that have undergone integrative aggregation. Our integrative aggregation methodology enables the acquisition of more discerning feature representations while preserving both semantic and spatial structural aspects, resulting in the estimation of highly accurate correspondences.

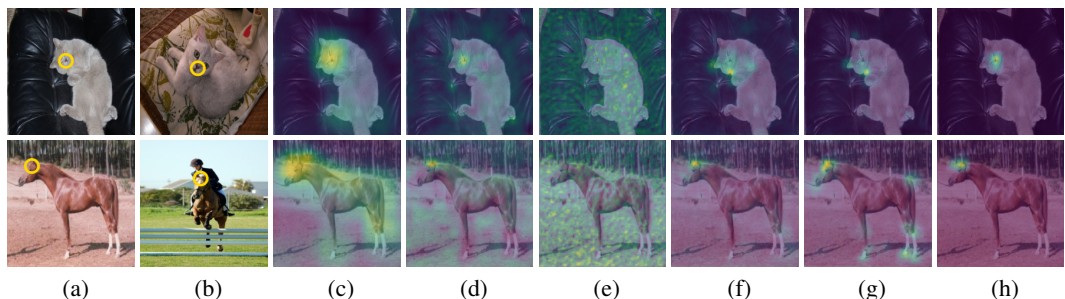

|     |     |     |     |     |     |     |     |
| (a) | (b) | (c) | (d) | (e) | (f) | (g) | (h) |

Figure 3: **Visualizations of cost volumes:** (a-b) source and target images. (c-e) **2D slices** of raw cost volumes at different levels $l$. (f) cost volumes constructed using feature maps that have undergone self-attention. (g) with feature maps that have undergone both self-attention and cross-attention, and (h) the cost volume that have undergone cost aggregation. From (c-e), the noises are suppressed in (h), while (f-g) shows aggregated features help to construct less noisy cost volumes. Note that the visualizations are obtained with respect to the circled point in the target image.

where the spatial structure is preserved in the features while sparse spatial locations with higher similarity to the query point is highlighted in the cost volume. Due to the different information encoded in their inputs, **the outputs of both aggregations have different characteristics.** In Fig. 2, compared to raw features in (c-d), the feature aggregation in (e-f) makes the features of semantic parts, *e.g.,* legs/claws and head, more consistent between the two birds. This is in agreement with observations in previous studies (Amir et al., 2021; Sun et al., 2021). On the other hand, compared to noisy cost volumes visualized in Fig. 3 (c-e), the aggregated cost volume in (h) is less noisy and more clearly highlights the most probable region for matching while less probable regions are suppressed. We additionally observe that **each type of aggregation can have apparent effects on the other.** Naturally, more robust descriptors can construct a less noisy cost volume as shown in Fig. 3 (f-g), and ease the subsequent cost aggregation process to promote more accurate correspondences. However, without a special model design, cost aggregation would not affect feature aggregation since it is performed after feature aggregation. Motivated by this, we proposed integrative aggregation, and its effects on feature maps are shown in Fig. 2 (g-h), where salient features become more distinguishable from each other, *i.e.,* the left and right claws. This exemplifies that more discriminative feature representations that are more focused and that also preserve semantics and spatial structure can be learned and reveals that potential benefits from their integration can be realized.

From these observations, we propose in the following a simple yet effective architecture that makes prudent use of both feature and cost aggregation.

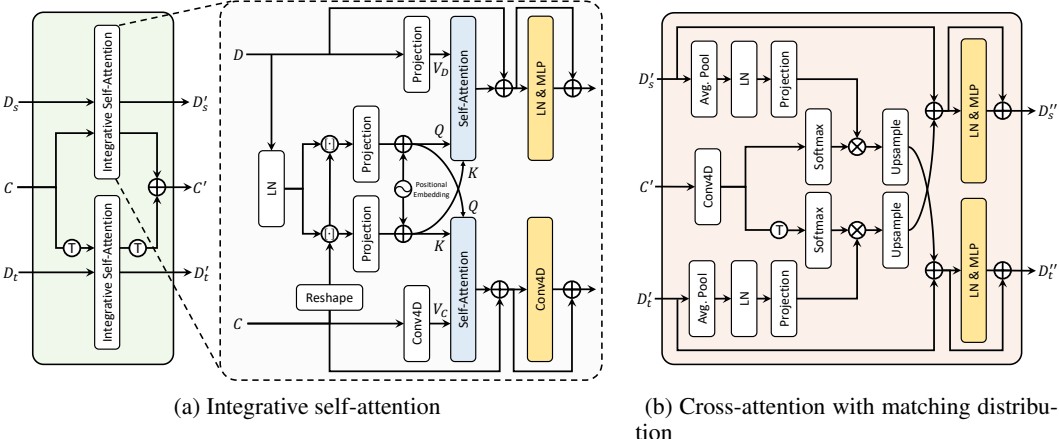

(a) Integrative self-attention

(b) Cross-attention with matching distribution

Figure 4: **Illustration of the proposed self- and cross-attention:** (a) joint feature aggregation and cost aggregation, and (b) cross-attention layer with matching distribution.

## 4 METHODOLOGY

### 4.1 PROBLEM FORMULATION

Let us first denote a pair of visually or semantically similar images, i.e., the source and target, as $I_s$ and $I_t$, the feature descriptors extracted from $I_s$ and $I_t$ as $D_s$ and $D_t$, respectively, and the cost volume computed between the feature maps as $C$. Given $I_s$ and $I_t$, we aim to establish a dense correspondence field $F(i)$ that is defined at all pixels $i$ and warps $I_s$ towards $I_t$. Given features extracted from deep CNNs (He et al., 2016) or Transformers (Dosovitskiy et al., 2020), we can construct and store a cost volume that consists of all pairwise feature similarities $C \in \mathbb{R}^{h \times w \times h \times w}$ with height $h$ and width $w$: $C(i,j) = D_s(i) \cdot D_t(j)$, where $i$ and $j$ index the source and target features, respectively. The dense correspondence field, $F(i)$, can then be determined from $C(i,j)$ considering all $j$.

### 4.2 PRELIMINARIES: SELF- AND CROSS-ATTENTION

We briefly explain the attention mechanism, a core component we extend from. Given a sequence of tokens as an input, Transformer (Vaswani et al., 2017) first linearly projects tokens to obtain query, key and value embeddings. These are then fed into a scaled dot product attention layer, followed by Layer Normalization (LN) (Ba et al., 2016) and a feed-forward network or MLP, to produce an output with the same shape as the input. Each token is attended to by all the other tokens. This projections are formulated as:

$$Q = \mathcal{P}_Q(X), \quad K = \mathcal{P}_K(X), \quad V = \mathcal{P}_V(X), \tag{1}$$

where $\mathcal{P}_Q$, $\mathcal{P}_K$ and $\mathcal{P}_V$ denote query, key and value projections, respectively, and $X$ denotes a token with a positional embedding. Subsequently, they pass through an attention layer:

$$\text{Attention}(X) = \text{softmax}(\frac{QK^T}{\sqrt{d_K}})V, \tag{2}$$

where $d_K$ is the dimension of the key embedding. Note that the $\text{Attention}(\cdot)$ function can be defined in various ways (Wang et al., 2020; Liu et al., 2021; Katharopoulos et al., 2020; Lu et al., 2021; Wu et al., 2021). Self- and cross-attention are distinguished by their input to the key and value projections. Given a pair of input tokens, e.g., $X_s$ and $X_t$, the input to the key and value projections of self-attention for $X_s$ is the same input, $X_s$, but for cross-attention across $X_s$ and $X_t$, the inputs to the key and value projection are $X_t$.

### 4.3 UNIFIED FEATURE AND COST AGGREGATION

**Integrative Self-Attention.** Toward more judicious use of both aggregations, we first leverage the fact that both feature descriptors $D_s$, $D_t$ and cost volume $C$ encode different information. To this end, in the proposed integrative self-attention layer, as shown in Fig. 4, we first obtain a feature cost volume $[D, C]$ by concatenating $D$ and $C$, where $[\cdot, \cdot]$ denotes concatenation.

This concatenation brings benefits from two perspectives. From the cost aggregation point of view, the feature map of the feature cost volume can disambiguate the initial noisy cost volume by referring to semantic-aware features as demonstrated in the stereo matching literature (Yoon & Kweon, 2006; Hosni et al., 2012; He et al., 2011), *i.e.,* cost volume filtering. From the feature aggregation point of view, the cost volume explicitly represents the similarity of features in one image with respect to the features in the other, and accounting for it drives the features in each image to become more compatible with those of the other. As the iterations unfold, this process will encourage both aggregations to benefit each other. In the end, the resultant feature representations will be more robust and discriminative as shown in Fig. 2 (g-h).

To compute self-attention, we take a different approach from other works (Sun et al., 2021; Cho et al., 2021) to define the query, key and value embeddings. Concretely, we define two independent value embeddings, specifically one for feature projection and the other for cost volume projection. Formally, we define the query, key and values as:

$$Q = \mathcal{P}_Q([D, C]), \quad K = \mathcal{P}_K([D, C]),$$
$$V_D = \mathcal{P}_{V_D}(D), \quad V_C = \mathcal{P}_{V_C}(C), \tag{3}$$

where $V_D$ and $V_C$ denote the value embeddings of feature descriptors and the cost volume, respectively. After computing an attention map by applying softmax over the query and key dot product, we use it to aggregate feature $D$ and cost volume $C$ with $V_D$ and $V_C$ using Eq. 2 as follows:

$$\text{Attention}_{\text{self-D}}(C, D) = \text{softmax}(\frac{QK^T}{\sqrt{d_K}})V_D,$$
$$\text{Attention}_{\text{self-C}}(C, D) = \text{softmax}(\frac{QK^T}{\sqrt{d_K}})V_C. \tag{4}$$

Note that any type of attention computation can be utilized, *i.e.,* additive (Bahdanau et al., 2014) or dot product (Vaswani et al., 2017), while in practice we use the linear kernel dot product with the associative property of matrix products (Katharopoulos et al., 2020). The outputs of this self-attention are denoted as $D'_s$, $D'_t$, and $C'$.

**Cross-Attention with Matching Distribution.** In the proposed cross-attention layer, the aggregated features and cost volume are explicitly used for further aggregation, and we condition both feature descriptors on both input images via this layer. By exploiting the outputs of the self-attention layer, the cross-attention layer performs cross-attention between feature descriptors for further feature aggregation using the improved feature descriptors $D'_s$, $D'_t$ and enhanced cost volume $C'$ from earlier aggregations.

As shown in Fig. 4, we first apply convolution to the input cost volume and treat the output as a cross-attention map, since applying a softmax function over the cost volume is tantamount to obtaining an attention map. In this way, an enhanced aggregation is enabled, as the input cost volume is transformed to represent a sharpened matching distribution. With a cross-attention map and value for the attention score defined as $QK^T = C'$ and $V_{D'} = \mathcal{P}_{V_D}(D')$, respectively, the subsequent attention process for cross-attention is then defined as follows:

$$\text{Attention}_{\text{cross}}(C', D') = \text{softmax}(\frac{C'}{\sqrt{d_K}})V_{D'}. \tag{5}$$

The outputs of this cross-attention are denoted as $D''_s$ and $D''_t$, and $C''$ is constructed using $D''_s$ and $D''_t$. The proposed attention layers are interleaved, and they are stacked $N$ times to facilitate the aggregations and increase the model capacity.

### 4.4 COARSE-TO-FINE FORMULATION

To improve the robustness of fine-scale estimates and enhance the benefits one aggregation gains from the other, we extend our architecture to a coarse-to-fine approach through pyramidal processing, as done in (Jeon et al., 2018; Melekhov et al., 2019; Truong et al., 2020b; Hong & Kim, 2021).

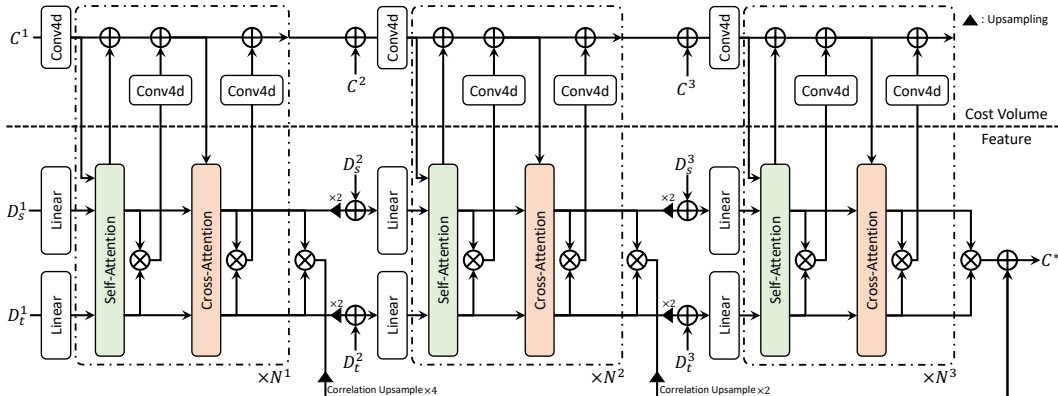

Figure 5: **Overall architecture of the proposed method.** Given feature maps $D_s$ and $D_t$ and the cost volume $C$ as inputs, our method employs self- and cross-attention specifically designed to conduct joint feature aggregation and cost aggregation in a coarse-to-fine manner.

We first use a coarse pair of refined feature maps and aggregated cost volume, and similar to (Zhao et al., 2021) that learns complementary correspondence by adding the cost volume of the previous scale, we progressively learn complementary descriptors and correspondences and encourage the coarser outputs to enhance the subsequent aggregations.

Formally, given the outputs of the attention block at each level, $D_s''^{,l}$, $D_t''^{,l}$ and $C''^{,l}$, where $l$ denotes the $l$-th level, we upsample the aggregated features using bilinear interpolation and add them to the raw feature descriptors extracted from $I_s$ and $I_t$ defined at the next level: $D_s^{l+1} = D_s^{l+1} + \mathrm{up}(D_s''^{,l})$, where $D_t^{l+1}$ is defined similarly. Note that we let the output cost volumes of self- and cross-attention at each level, $C'^{,l}$ and $C''^{,l}$, undergo convolution and residual connections, *i.e.*, $C''^{,l} = C'^{,l} + \mathrm{Conv4d}(C''^{,l})$, to facilitate the training process. Then, we define the next-level cost volume as $C^{l+1} = C^{l+1} + C''^{,l}$.

Finally, given the features $D_s''$ and $D_t''$ at each level, we compute the cost volume, and the sum of all cost volumes across all levels are added up to obtain the final output $C^*$ that is used to estimate the final flow field, as shown in the bottom of Fig. 5. We empirically find that increasing $N$ and $l$ appreciably boosts correspondence performance. However, there also exists a speed-accuracy trade-off of choosing $N$ and $l$, for which we provide an analysis in the supplementary material.

### 4.5 INFERENCE: DENSE ZOOM-IN

At the inference phase, we leverage multi-scale predictions to predict highly accurate correspondences. The goals of this approach are two-fold: to prevent a large memory increase when processing high-resolution input image pairs, *e.g.,* HD or Full HD, and to capture possible fine-grained correspondences missed by the coarse-to-fine design.

Dense zoom-in consists of three stages. For the first stage, UFC takes an input image pair and uses the output flow to coarsely align the source image to the target image as similarly done in (Shen et al., 2020). Unlike RANSAC-Flow (Shen et al., 2020), we do not resort to finding a homography transformation, but rather rely on the output flow itself. We empirically find that for images with extreme geometric deformations, reliable homography transformations may not be found.

Subsequently, we evenly partition the coarsely aligned source image and the target image into $k \times k$ local windows, where $k$ is a hyperparameter. Each pair of partitioned local windows at the same location is then used to find more fine-grained correspondences by feeding them into UFC to obtain local flow fields. Note that in this stage, we also compute a cycle-consistency confidence score Jiang et al. (2021) that will be used in the final decision-making process. To enable multi-scale inference, we choose multiple $k$, *e.g.,* $k = 2$, for which we provide an ablation study in the supplementary material. We then perform transitive composition Zhou et al. (2016) using the coarse flow and each of the multi-scale flows. Finally, using the confidence values of composited flows at each pixel, we

| Methods | Train Image Reso. | Keypoint Annotation Reso. | Feat. Agg. | Cost Agg. | SPair-71k PCK @ $\alpha_{bbox}$ | | | | | PF-PASCAL PCK @ $\alpha_{img}$ | | | PF-WILLOW PCK @ $\alpha_{bbox}$ | | PCK @ $\alpha_{bbox-kp}$ | |
|---|---|---|---|---|---|---|---|---|---|---|---|---|---|---|---|---|
| | | | | | 0.01 | 0.03 | 0.05 | 0.1 | 0.15 | 0.05 | 0.1 | 0.15 | 0.05 | 0.1 | 0.05 | 0.1 |
| DHPF | 240×240 | 240 | - | RHM | 1.74 | 11.0 | 20.9 | 37.3 | 47.5 | 75.7 | 90.7 | 95.0 | 49.5 | 77.6 | - | 71.0 |
| SCOT | - | Max 300 | | OT-RHM | - | - | - | 35.6 | - | 63.1 | 85.4 | 92.7 | - | - | 47.8 | 76.0 |
| CHM | 240×240 | 240 | 2D Conv. | 6D Conv. | 2.25 | 14.9 | 27.2 | 46.3 | 57.5 | 80.1 | 91.6 | 94.9 | 52.7 | 79.4 | - | 69.6 |
| CATs | 256×256 | 256 | - | Trans. | 1.90 | 13.8 | 27.7 | 49.9 | 61.7 | 75.4 | 92.6 | 96.4 | 50.3 | 79.2 | 40.7 | 69.0 |
| MMNet-FCN | 224×320 | 224×320 | Conv. + Trans. | - | 2.80 | 18.8 | 33.3 | 50.4 | 61.2 | 81.1 | 91.6 | 95.9 | - | - | - | - |
| PWarpC-NC-Net | 400×400 | Ori | - | 4D Conv. | 2.55 | 17.1 | 31.6 | 52.0 | 61.8 | 79.2 | 92.1 | 95.6 | - | - | 48.0 | 76.2 |
| SCorrSAN | 256×256 | 256 | Linear | - | - | - | - | 55.3 | - | 81.5 | 93.3 | 96.6 | 54.1 | 80.0 | - | - |
| VAT | 512×512 | 512 | - | 4D Conv. + Trans. | 3.17 | 19.6 | 35.0 | 55.5 | 65.1 | 78.2 | 92.3 | 96.2 | 52.8 | 81.6 | 42.3 | 71.3 |
| CATs++ | 512×512 | 512 | - | 4D Conv. + Trans. | 4.31 | 25.0 | 40.7 | 59.8 | 68.5 | 84.9 | 93.8 | 96.8 | 56.7 | 81.2 | 47.0 | 72.6 |
| TransforMatcher | 240×240 | 240 | - | Trans. | - | - | - | 53.7 | - | 80.8 | 91.8 | - | - | 76.0 | - | 65.3 |
| NeMF | 512×512 | Ori | - | 4D Conv. + Trans. | 3.2 | 19.5 | 34.2 | 53.6 | - | 80.6 | 93.6 | - | - | - | 60.8 | 75.0 |
| **UFC** | 512×512 | Ori | Integrative Transformer | | **8.40** | **34.1** | **48.5** | **64.4** | **72.1** | **88.0** | **94.8** | **97.9** | **58.6** | 81.2 | **50.4** | 74.2 |

Table 1: **Semantic matching results.**

| Methods | Feat.Agg. | Cost.Agg. | HPatches Original AEPE ↓ | | | | | | PCK ↑ | ETH3D AEPE ↓ | | | | | | | |
|---|---|---|---|---|---|---|---|---|---|---|---|---|---|---|---|---|---|
| | | | I | II | III | IV | V | Avg. | 5px | rate=3 | rate=5 | rate=7 | rate=9 | rate=11 | rate=13 | rate=15 | Avg. |
| COTR | Trans. | - | - | - | - | - | - | 7.75 | 91.10 | 1.66 | 1.82 | 1.97 | 2.13 | 2.27 | 2.41 | 2.61 | 2.12 |
| PUMP | - | 4D Conv. | - | - | - | - | - | 2.87 | 97.14 | 1.77 | 2.81 | 2.39 | 2.39 | 3.56 | 3.87 | 4.57 | 3.05 |
| ECO-TR | Trans. | - | - | - | - | - | - | 2.52 | 90.85 | 1.48 | 1.61 | 1.72 | 1.89 | 1.97 | 2.06 | 2.18 | 1.87 |
| COTR+Interp. | Trans. | - | - | - | - | - | - | 7.98 | 86.33 | 1.71 | 1.92 | 2.16 | 2.47 | 2.85 | 3.23 | 3.76 | 2.59 |
| **UFC + (C)** | Integrative Transformer | | **0.87** | **1.29** | **1.37** | **3.19** | **1.92** | **1.73** | **98.76** | **1.45** | **1.59** | **1.64** | 1.76 | 1.82 | 1.90 | 1.95 | **1.73** |
| GLU-Net | - | 2D Conv. | 1.55 | 12.66 | 27.54 | 32.04 | 52.47 | 25.05 | 78.54 | 1.98 | 2.54 | 3.49 | 4.24 | 5.61 | 7.55 | 10.78 | 5.17 |
| GLU-Net-GOCor | - | Hand-crafted | 1.29 | 10.07 | 23.86 | 27.17 | 38.41 | 20.16 | 81.43 | 1.93 | 2.28 | 2.64 | 3.01 | 3.62 | 4.79 | 7.80 | 3.72 |
| DMP | 2D Conv. | 2D Conv. | 3.21 | 15.54 | 32.54 | 38.62 | 63.43 | 30.64 | 63.21 | 2.43 | 3.31 | 4.41 | 5.56 | 6.93 | 9.55 | 14.20 | 6.62 |
| PDCNet (MS) | - | 2D Conv. | **1.15** | 7.43 | 11.64 | 25.00 | 30.49 | 15.14 | **91.41** | 1.60 | 1.79 | 2.00 | 2.26 | 2.57 | 2.90 | 3.56 | 2.38 |
| GMFlow | Trans. | - | 4.72 | 26.46 | 40.75 | 62.49 | 79.80 | 42.85 | 69.50 | 1.64 | 1.86 | 2.12 | 2.36 | 3.49 | 5.62 | 10.64 | 3.96 |
| COTR† | Trans. | - | 19.65 | 33.81 | 45.81 | 62.03 | 66.28 | 45.52 | 5.10 | 8.76 | 9.86 | 11.23 | 12.44 | 13.77 | 14.94 | 16.09 | 12.44 |
| PDC-Net+ (MS) | - | 2D Conv. | - | - | - | - | - | - | - | 1.58 | 1.76 | 1.96 | 2.16 | 2.49 | 2.73 | 3.24 | 2.27 |
| **UFC** | Integrative Transformer | | 1.91 | **6.13** | **5.62** | **6.36** | **19.44** | **7.88** | 89.36 | **1.54** | **1.72** | **1.99** | 2.18 | 2.58 | 2.66 | 3.01 | 2.24 |

Table 2: **Geometric matching results.** A higher scene label or rate, *i.e.,* V or 15, consists of more difficult images with extreme geometric deformations. † : Dense evaluation without zoom-in technique and confidence thresholding. *(C) : Confidence thresholding.*

select the flow with the highest confidence score. This selection is performed for every pixel and results in a final dense flow map.

## 5 EXPERIMENTS

### 5.1 SEMANTIC MATCHING

We first evaluate ours on semantic matching, where large intra-class variations and background clutters pose additional challenges to matching. We use three standard benchmarks: SPair-71k (Min et al., 2019b), PF-PASCAL (Ham et al., 2017) and PF-WILLOW (Ham et al., 2016). We follow the evaluation protocol of (Cho et al., 2022b). The results are summarized in Table 1, where UFC outperforms others for all the benchmarks at almost all PCKs, showing robustness to the above challenges. While VAT (Hong et al., 2022a) and NeMF (Hong et al., 2022b) perform better at $\alpha_{bbox}$, we stress that VAT evaluates at higher resolution and NeMF specializes in fine-grained correspondences. Also, we note that our performance is slightly inferior to SCOT (Liu et al., 2020) and PWarPC (Truong et al., 2022) for $\alpha_{bbox-kp} = 0.1$, but this is compensated for by the superior performance at lower alpha, *i.e.,* 0.05, and the fact that PF-PASCAL (Ham et al., 2017) and PF-WILLOW (Ham et al., 2016) are small-scale datasets with a limited number of image pairs. Moreover, we highlight that for SPair-71k (Min et al., 2019b), the largest dataset in semantic correspondence with extreme viewpoint and scale difference, UFC outperforms competitors at all PCKs.

### 5.2 GEOMETRIC MATCHING

We next show that our method also performs very well in geometric matching. Following the evaluation protocol of (Truong et al., 2021), we report the results on HPatches (Balntas et al., 2017) and ETH3D (Schops et al., 2017) in Table 2. From the results, our method clearly outperforms existing dense matching networks, including those that perform additional optimization (Truong et al., 2020a; Hong & Kim, 2021; Truong et al., 2021) and inference strategies (Jiang et al., 2021; Truong et al., 2021), and a representative optical flow method, GMFlow (Xu et al., 2021). Note that UFC excels at finding correspondences under extreme geometric deformations, *i.e.,* scene IV and V. Interestingly, as UFC outperforms others at all intervals of ETH3D that consist of image sequences with varying magnitudes of geometric transformations, this indicates that it can also perform well in optical flow settings. This is supported by the results of GMFlow (Xu et al., 2021), where we consistently achieve better performance. To ensure a fair comparison to COTR (Jiang et al., 2021) and its follow-up works (Revaud et al., 2022; Tan et al., 2022), we present results from a variant of

our method, denoted as (C). Moreover, we also include COTR† to represent a truly dense version of COTR (Jiang et al., 2021), and we observe that UFC clearly performs better.

## 5.3 QUANTITATIVE COMPARISON BETWEEN AGGREGATION STRATEGIES

Figures 2 and 3 qualitatively show that the information both aggregations exploit and the output they generate differ. Here, we empirically show that these lead to appreciable performance differences. Table 3 (I-III) compares different aggregation strategies that are trained and evaluated on both tasks. Note that for these

|       |                                            | HPatches | SPair-71k |
|-------|--------------------------------------------|----------|-----------|
|       |                                            | AEPE $\downarrow$ | $\alpha_{bbox} = 0.1 \uparrow$ |
| **(I)**   | Feature self-att.                      | 78.2     | 36.1      |
| **(II)**  | Feature self-att. and cross-att.       | 59.9     | 38.5      |
| **(III)** | Cost self-att.                         | 51.8     | 33.6      |
| **(IV)**  | Feature self-att. + cost self-att.     | 36.8     | 51.7      |
| **(V)**   | Feature self- and cross-att. + cost self-att. | 26.2 | 56.5 |

Table 3: **Comparison of aggregation strategies.**

variants, we do not include any module for boosting performance, e.g., coarse-to-fine, multi-level or multi-scale features, and maintain a similar number of learnable parameters. Pytorch-like pseudocodes and additional visualizations are given in the supplementary material. From the results, we find that each aggregation strategy yields apparently different results in two tasks as reported in (I-III). A particularly illustrative comparison is (I) vs. (III), where only self-attention is performed on features or the cost volume, which clearly differentiate them. As expected, we also observe that performing cross-attention improves performance.

In the last two rows, we report the results of variants that utilize both feature and cost aggregation. (V) is our integrative aggregation, and (IV) is a naïve sequential aggregation where feature aggregation is followed by cost aggregation. They both achieve large performance boosts, as expected since the resultant cost volume in Fig. 3 (f) and (h) includes less noisy and more concentrated scores compared to (c-e), while integrative aggregation clearly performs better. From these quantitative comparisons, we highlight that the two types of aggregation serve different purposes that lead to apparent performance differences, and the potential benefits arising from their relationship can be further exploited with our proposed design.

## 5.4 ABLATION STUDY

In this ablation study, we verify the need for each component of our method. Table 4 presents quantitative results of each variant on both geometric and semantic matching. The baseline represents a variant equipped with self- and cross-attention on feature maps. From (II) to (VI), each proposed component is pro-

|       | Components | HPatches | SPair-71k |
|-------|-----------|----------|-----------|
|       |           | AEPE $\downarrow$ | $\alpha_{bbox} = 0.1 \uparrow$ |
| **(I)**   | Baseline                          | 36.8  | 51.7  |
| **(II)**  | Integrative self-att.             | 32.8  | 54.7  |
| **(III)** | Integrative self- and cross-att.  | 22.8  | 58.4  |
| **(IV)**  | + matching distribution.          | 18.7  | 59.9  |
| **(V)**   | + hierarchical processing         | 10.9  | 64.4  |
| **(VI)**  | + dense zoom-in                   | 7.88  | 64.4  |

Table 4: **Component ablation study.**

gressively included. We control the number of learnable parameters for each variant to be similar.

Comparing (I) and (II), we find that the proposed integrative self-attention layer benefits from joint aggregation of features and cost volume, achieving improved performance on both tasks. We next find that each component clearly helps to boost performance. Interestingly, we find dramatic improvements when cross-attention is included (III), indicating that explicit conditioning between input images is helpful. This further enhanced by using the aggregated cost volume as a cross-attention map (IV). We empirically find that our dense zoom-in brings marginal gain for semantic matching. We believe the sparse keypoint evaluation metric and the relatively small resolution images of the benchmark disturb harnessing the advantages of our inference strategy.

## 6 CONCLUSION

In this paper, we introduced a simple yet effective dense matching approach, Unified Feature and Cost Aggregation with Transformers (UFC), that capitalizes on the distinct advantages of the two types of aggregation. We further devise an enhanced aggregation through cross-attention with matching distribution. This method is formulated in a coarse-to-fine manner, yielding an appreciable performance boost. We have shown that our approach exhibits high speed and efficiency and that it surpasses all other existing works on several benchmarks, establishing new state-of-the-art performance.

ACKNOWLEDGEMENT

This research was supported by the MSIT, Korea (IITP-2024-2020-0-01819, ICT Creative Consilience Program, RS-2023-00227592, Development of 3D Object Identification Technology Robust to Viewpoint Changes), and National Research Foundation of Korea (NRF-2021R1A6A1A03045425).

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
