# Unifying Feature and Cost Aggregation with Transformers for Semantic and Visual Correspondence

**Sunghwan Hong**[*], **Seokju Cho**[*], **Seungryong Kim**[††]
Korea University
{sung_hwan,seokju_cho,seungryong_kim}@korea.ac.kr

**Stephen Lin**
Microsoft Research Asia
stevelin@microsoft.com

## Overview

In the following, we first provide more implementation details in Section A. Then, we provide details on evaluation metrics and datasets in Section B. We then explain the training procedure in more depth in Section C. Subsequently, we provide additional experimental results and ablation study in Section D. We then provide clarifications to the evaluation procedure adopted by COTR (Jiang et al., 2021) and its follow-up works in Section E. We also provide Pytorch-Like psuedo-codes in Section F. Finally, we present qualitative results for all the benchmarks in Section G and a discussion of future work in Section H.

## A    Implementation Details

### A.1    Network Architectures

To extract features, we use ResNet-101 (He et al., 2016) for semantic matching, and VGG-16 (Simonyan & Zisserman, 2014) for geometric matching, consistent with prior works (Min et al., 2020; 2021; Cho et al., 2021; Hong et al., 2022a; Truong et al., 2020b;a; 2021b). We select three feature maps from the last convolutional block, namely $\mathrm{Conv3\_x}$, $\mathrm{Conv4\_x}$, and $\mathrm{Conv5\_x}$, with channel dimensions of 2048, 1024, and 512, respectively. To reduce computational complexity, we project each feature map to smaller dimensions of 384, 256, and 128, respectively, before constructing a cost volume and passing it to our integrative aggregation block. Bilinear interpolation is used to adjust the spatial dimensions of intermediate outputs. The resolutions at each levels $l = 1, 2, 3$ are $16 \times 16$, $32 \times 32$ and $64 \times 64$, respectively. For the final output flow map, we use a soft-argmax operator with temperature set to 0.02.

### A.2    Other Implementation Details

**COTR Implementation Details.**    In the main table, we report the results of COTR (Jiang et al., 2021) without zoom-in and confidence thresholding. Here, we provide the implementation details for how we obtained the results.

To adapt the input pair of images for use in our evaluation, we resize them to $256 \times 256$, which matches the resolution used by COTR (Jiang et al., 2021). Rather than selecting sparse coordinates for finding correspondences, we input all coordinates defined at the original resolution of the images, resulting in dense correspondences. This approach involves feed-forwarding all coordinates in parallel, which speeds up the process. We then compute the average endpoint error (AEPE) for all correspondences, masking any invalid correspondences as per conventional evaluation protocols (Truong et al., 2020b;a; 2021b). Note that our evaluation does not take into account the zoom-in

---

[*]equal contribution
[††]Corresponding author

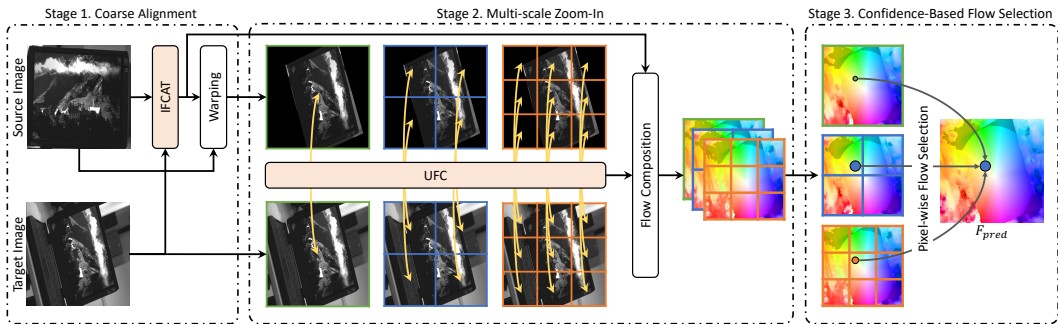

Figure 1: **Pipeline of dense zoom-in technique.**

technique used in COTR. We also evaluate dense correspondence rather than the original sparse or quasi-dense evaluation method adopted by COTR (Jiang et al., 2021), which we detail in Section E.

**GMFlow Inference Details.** To evaluate GMFlow on HPatches (Balntas et al., 2017) and ETH3D (Zhao et al., 2021), we use the weights given at the official implementation page. However, processing HPatches at its original resolution requires an enormous amount of memory, even when using a high-end GPU such as the 80GB A100. Therefore, we interpolate the input image pairs using bilinear interpolation to match the size of the crops used during training ($320 \times 896$), before feeding them into GMFlow (Xu et al., 2021). We use the weights trained on Sintel (Butler et al., 2012) with refinement strategy to obtain the best results.

Since there are numerous hyperparameters that can affect model performance, *i.e.,* padding factor, window sizes for Swin Transformer (Liu et al., 2021) and scale factor, we choose the same hyperparameters $u$ that were used to obtain the pre-trained weight. Specifically, we set padding factor to 32, upsample factor to 4, scale factor to 2, attention split list to (2,8), correlation radius list to (-1,4) and flow propagation list to (-1,1). We keep all other hyperparameters at their default values. We then evaluate the model using the same evaluation procedure as in previous works (Truong et al., 2020b;a; 2021b).

**PDC-Net Inference Details.** To evaluate PDC-Net (Truong et al., 2021b) on HPatches (Balntas et al., 2017) and ETH3D (Schops et al., 2017), we simply use the pre-trained weights and the official implementation codes. Note that the only hyperparameters we change at inference are GO-Cor (Truong et al., 2020a) hyperparameters, for which we set the number of iterations for global and local correlation map optimization to 3 and 7, respectively. The rest of the hyperparameters remain as the default values.

## A.3 DENSE ZOOM-IN

We provide an overview of dense zoom-in used at inference phase, in Fig. 1.

## B EVALUATION METRICS AND DATASETS

### B.1 GEOMETRIC MATCHING

**Compared Methods.** There are two groups of methods we compare to. The first group includes COTR (Jiang et al., 2021), PUMP (Revaud et al., 2022) and ECO-TR (Tan et al., 2022). The second group includes GLU-Net (Truong et al., 2020b), GOCor (Truong et al., 2020a), DMP (Hong & Kim, 2021), PDC-Net (Truong et al., 2021b), PDC-Net+ (Truong et al., 2021a), and they are trained in a self-supervised manner on either DPED-CityScape-ADE (Ignatov et al., 2017; Cordts et al., 2016; Zhou et al., 2019) or MegaDepth (Li & Snavely, 2018) except for GMFlow (Xu et al., 2021). For the second group, the evaluation is done using https://github.com/PruneTruong/DenseMatching.

**Evaluation Metric.** For the evaluation metric, we use the average end-point error (AEPE), computed by averaging the Euclidean distance between the ground-truth and estimated flow, and percentage of correct keypoints (PCK), computed as the ratio of estimated keypoints within a threshold

of the ground truth to the total number of keypoints. More specifically, AEPE is computed using the following equation: $\|F_{\mathrm{GT}} - F_{\mathrm{pred}}\|_2$, where $F_{\mathrm{GT}}$ represents a ground-truth dense flow map and $F_{\mathrm{pred}}$ represents a dense predicted flow map.

**HPatches.** Hpatches (Balntas et al., 2017) consists of images with different views of the same scenes. Each sequence contains a source and five target images with different viewpoints and the corresponding ground-truth flows. Generally, the later scenes, *i.e.,* IV and V, consist of more challenging target images. We use images of high resolutions ranging from $450 \times 600$ to $1,613 \times 1,210$.

**ETH3D** Unlike Hpatches (Balntas et al., 2017), ETH3D (Schops et al., 2017) consists of real 3D scenes, where the image transformations are not constrained to a homography. This multi-view dataset contains 10 image sequences ranging from $480 \times 752$ to $514 \times 955$. The authors of ETH3D (Schops et al., 2017) additionally provide a set of sparse image correspondences, for which we follow the protocol of (Truong et al., 2020b) by sampling the image pairs at different intervals to evaluate on varying magnitudes of geometric transformations. We evaluate on 7 intervals in total, each interval containing approximately 500 image pairs, or 600K to 1000K correspondences. Generally, the image pairs are more challenging at a higher rate, *i.e.,* rate 13 or 15, as shown in Fig. 5 and Fig. 6.

## B.2 SEMANTIC MATCHING

**Compared Methods.** We compare our methods to semantic matching methods, which are all trained in a supervised manner using ground-truth keypoints. DHPF (Min et al., 2020), SCOT (Liu et al., 2020), CHM (Min et al., 2021), CATs (Cho et al., 2021), MMNet (Zhao et al., 2021), PWarpC-NC-Net (Truong et al., 2022; Rocco et al., 2018), SCorrSAN (Huang et al., 2022), VAT (Hong et al., 2022a), CATs++ (Cho et al., 2022b), TransforMatcher (Kim et al., 2022) and NeMF (Hong et al., 2022b) are trained with SPair-71k (Min et al., 2019b) when evaluated on SPair-71k and they are trained on PF-PASCAL (Ham et al., 2017) when evaluated on PF-PASCAL and PF-WILLOW (Ham et al., 2016). Note that all the methods adopt ResNet-101 except for MMNet (Zhao et al., 2021).

**Evaluation Metric.** For semantic matching, following (Rocco et al., 2018; Min et al., 2019a; 2020; Cho et al., 2021; 2022a), we transfer the annotated keypoints in the source image to the target image using the dense correspondence between the two images. The percentage of correct keypoints (PCK) is computed for evaluation. Note that higher PCK values are better. Concretely, given predicted keypoint $k_{\mathrm{pred}}$ and ground-truth keypoint $k_{\mathrm{GT}}$, we count the number of predicted keypoints that satisfy the following condition: $d(k_{\mathrm{pred}}, k_{\mathrm{GT}}) \leq \alpha \cdot \max(H, W)$, where $d(\cdot)$ denotes Euclidean distance; $\alpha$ denotes a threshold value; $H$ and $W$ denote height and width of the object bounding box or the entire image. Note that we additionally reported results of PCK @ $\alpha_{\mathrm{bbox\text{-}kp}}$ to compensate for the fact that (Min et al., 2020; 2021; Cho et al., 2021; Hong et al., 2022a; Huang et al., 2022) chose thresholds different from other works (Liu et al., 2020; Lee et al., 2019; Rocco et al., 2018; 2017) for PF-WILLOW (Ham et al., 2016).

**SPair-71k.** SPair-71k (Min et al., 2019b) is a large-scale benchmark for semantic correspondence, which consists of 18 object categories of 70,958 image pairs with extreme and diverse viewpoints, scale variations, and rich annotations for each image pair. Ground-truth annotations for object bounding boxes, segmentation masks and keypoints are available. For the evaluation, we follow the conventional evaluation protocol (Min et al., 2019a) of using a test split of 12,234 image pairs.

**PF-PASCAL and PF-WILLOW** PF-PASCAL (Ham et al., 2017) is a dataset introduced as an extension for PF-WILLOW (Ham et al., 2016). It consists of 1,351 image pairs of 20 image categories, while PF-WILLOW (Ham et al., 2016) consists of 900 image pairs of 4 image categories. PF-PASCAL (Ham et al., 2017) is a more challenging dataset than others, *i.e.,* TSS (Taniai et al., 2016) or PF-WILLOW (Ham et al., 2016), for semantic correspondence evaluation, as it additionally exhibits large appearance, scene layout, scale and clutter changes. For evaluation, we use the test split of PF-PASCAL (Ham et al., 2017) and PF-WILLOW (Ham et al., 2016).

## C TRAINING DETAILS

In this section, we provide training details for both semantic and geometric matching. We employ an Intel Core i7-10700 CPU and RTX-3090 GPUs for training.

## C.1 Dense Geometric Matching

For geometric matching, we adopt two-stage training. At the first stage, we freeze the backbone network and only train UFC using DPED-CityScape-ADE (Ignatov et al., 2017; Cordts et al., 2016; Zhou et al., 2019). This stage is similar to the training procedure of GOCor-GLU-Net (Truong et al., 2020a). More specifically, due to the limited amount of dense correspondence data, most matching networks resort to self-supervised training, where synthetic warps provide dense correspondences. To this end, we adopt the same training procedure to GOCor-GLU-Net (Truong et al., 2020a) that consists of pairs of images created by synthetically warping the image according to random affine, homography or TPS transformations. We crop the images to $512 \times 512$, and in total, we use 40K image pairs for the first stage of training. We set the learning rate to $3e^{-4}$, use AdamW (Loshchilov & Hutter, 2017) and iterate for 50 epochs with the batch size set to 16. We freeze the backbone in this stage.

For the second stage, we continue from the best model from the first stage, which was chosen by cross-validation. For the dataset, we use the MegaDepth dataset, which consists of 196 different scenes reconstructed from about 1M internet images using COLMAP (Schonberger & Frahm, 2016) and combine this with the synthetic data. For training, we sample up to 500 random images from 150 different scenes in which the overlap is at least 30% with the sparse SfM point cloud. We also include random independently moving objects sampled from the COCO (Lin et al., 2014) dataset on top of the synthetic data. Moreover, we also utilize perturbation data as we found it beneficial to include in the dataset. Finally, for the validation dataset, we sample up to 80 random images pairs from 25 different scenes. We resize the images to $512 \times 512$ in consistency with the first stage. For the second stage, we train the whole network, set the learning rate to $1e^{-4}$, use AdamW (Loshchilov & Hutter, 2017) and iterate for 175 epochs with the batch size set to 16.

**Dense Semantic Matching** To ensure a fair comparison, following (Min et al., 2021; Cho et al., 2021), when evaluating on SPair-71k (Min et al., 2019b) we train the proposed method on the training split of SPair-71k (Min et al., 2019b), and when evaluating on PF-PASCAL (Ham et al., 2017) and PF-WILLOW (Ham et al., 2016) we train on the training split of PF-PASCAL (Ham et al., 2017). We only train the UFC module and freeze the backbone network. We apply random augmentation (Buslaev et al., 2020) as done in (Cho et al., 2021). We set the learning rate to $1e^{-4}$, use AdamW (Loshchilov & Hutter, 2017) as an optimizer, set the batch size to 24, and iterate for 50 epochs for SPair-71k (Min et al., 2019b) and 300 epochs for PF-PASCAL (Ham et al., 2017). The best model is obtained through cross-validation.

# D  Additional Quantitative Results and Ablation Study

**Computational complexity.** Here, we examine the computational complexity of the proposed approach relative to other methods. In Table 1, we report memory consumption, run time and number of learnable parameters. Note that memory consumption is in Gigabytes measuring maximum GPU memory utilization at the inference phase, run-time is in seconds, and the number of learnable parameters is in Millions.

| | Methods | Memory [GB] | Run-time [s] | # of param. [M] |
|---|---|---|---|---|
| HPatches | COTR | 5.7 | up to 216.0 | 18.5 |
| | GMFlow | OOM (80.0 <) | OOM | 3.5 |
| | GMFlow [1] | 17.8 | 0.12 | 3.5 |
| | PDC-Net (D) | 2.4 | 1.72 | 10.8 |
| | PDC-Net+ (MS) | OOM(24.0 <) | OOM | 10.8 |
| | **UFC**$_{geometric}$ | 4.9 | 3.1 | 15.5 |
| SPair-71k | MMNet | 4.9 | 0.17 | 251.3 |
| | CATs | 1.6 | 0.05 | 4.6 |
| | VAT | 3.8 | 0.19 | 3.3 |
| | CATs++ | 3.1 | 0.26 | 5.5 |
| | **UFC**$_{semantic}$ | 3.5 | 0.17 | 14.5 |

Table 1: **Efficiency comparison.**

In Table 1, UFC is able to consolidate large resolution images from HPatches (Balntas et al., 2017), *i.e.,* $1{,}613 \times 1{,}210$, with only 4.9 GB memory consumption, which is a large saving compared to competitors (Xu et al., 2021; Truong et al., 2021a; Jiang et al., 2021). For inference time per image pair, UFC runs 2x faster than PDC-Net, while approximately 100x faster than COTR. The overall efficiency of UFC stems from: adjusted the resolution of the intermediate cost volumes, $C^l$, to the coarsest resolution within the coarse-to-fine design, efficient attention computation, adequate hyperparameters set for $N, k, l$ and finally, compared to (Zhao et al., 2021; Cho et al., 2021; 2022b; Hong et al., 2022a) that use a large number of feature maps, typically leveraging all the feature maps from conv2_x to conv5_x, UFC only uses 3 feature maps, each used for each level $l$.

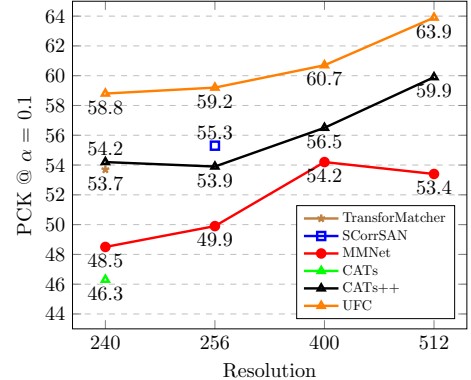

Figure 2: **Image resolution ablation.**

| | N | AEPE | Memory [GB] | Run-time [ms] | # of param. [M] |
|---|---|---|---|---|---|
| Hpatches | (1,0,0) | 24.8 | 1.3 | 117.6 | 5.7 |
| | (2,0,0) | 18.7 | 1.4 | 141.6 | 10.4 |
| | (1,1,1) | 18.2 | 2.5 | 158.6 | 8.3 |
| | (2,2,1) | 13.2 | 2.6 | 193.2 | 14.5 |
| | (2,2,2) | 10.9 | 2.9 | 200.2 | 15.5 |
| | (3,3,3) | 10.1 | 3.6 | 269.1 | 22.6 |
| | (1,1,1,1) | 12.1 | 13.4 | 162.3 | 8.73 |
| | N | PCK | Memory [GB] | Run-time [ms] | # of param. [M] |
| SPair-71k | (1,0,0) | 55.4 | 1.6 | 89.8 | 5.7 |
| | (2,0,0) | 59.9 | 1.7 | 120.8 | 10.4 |
| | (1,1,1) | 63.7 | 2.8 | 142.2 | 8.3 |
| | (2,2,1) | 64.4 | 3.5 | 168.6 | 14.5 |
| | (2,2,2) | 62.5 | 3.9 | 189.4 | 15.5 |
| | (3,3,3) | 59.8 | 4.6 | 237.4 | 22.6 |
| | (1,1,1,1) | 63.1 | 14.4 | 171.5 | 8.85 |

Table 2: **Depth ablation.**

**Ablation study on image resolution**   Here, we show additional results of our method UFC trained and evaluated at different resolutions on SPair-71k (Min et al., 2019b). As done in CATs++ (Cho et al., 2022a), we train and evaluate at 240, 256, 400 and 512 to directly compare with competitors, each of which train and evaluate at different resolutions, *i.e.,* 240 for CHM (Min et al., 2021), 256 for CATs (Cho et al., 2021), 400 for PMNC (Lee et al., 2021) and 512 for CATs++ (Cho et al., 2022a). The results are shown in Fig. 2, where UFC still outperforms the competitors.

**Ablation study on $N$ and $l$.**   We show how the performance, memory consumption, run time and the number of learnable parameters vary as we set different $N$ and $l$. The measured units are similar to Tab. 2, but run-time is instead measured in milliseconds, since zoom-in is excluded for these results. The results are shown in Table 2. We design our model that for each level $l$, a user can freely set the depth similar to other Transformers (Vaswani et al., 2017; Liu et al., 2021). Simply, if we increase the depth, this will lead to increasing the capacity of our model. Note that when the value of $l$ exceeds 3, a more efficient approach is to avoid computing the entire cost volume $C^4$. If $C^4$ were explicitly computed, it would have been used as a residual. However, in this case, we choose not to use it as a residual. This way, we can extend our architecture to process at higher resolution without introducing large memory consumption.

The effects of varying $N$ are that generally, we observe apparent improvements in performance as it is increased, but as the model gets heavier, it will face slower inference time and increased memory consumption. Also, it is notable that the number of learnable parameters can have significant effects on the memory consumption for training and the generalization power, which can be a limitation if the model capacity is increased. To alleviate this, we choose to sacrifice some performance for greater efficiency. The depths can be controlled easily by users depending on whether they prefer efficiency or performance. However, it should be noted that increases in depths will not always have a positive influence on the performance as shown for the semantic matching results. We suspect that the inferior results are caused by the overfitting problem and the lack of data in the semantic

| k | AEPE | | Memory | Run-time |
|---|---|---|---|---|
| | HPatches (Balntas et al., 2017) | ETH3D (Schops et al., 2017) | [GB] | [s] |
| $(2, 3)$ | 7.79 | 2.57 | 4.9 | 3.13 |
| $(2, 3, 4)$ | 7.90 | 2.49 | 6.2 | 6.05 |
| $(3, 4)$ | 7.89 | 2.47 | 6.0 | 5.32 |
| $(3, 4, 5)$ | 7.88 | 2.33 | 14.8 | 4.63 |
| $(4, 5)$ | 7.99 | 2.42 | 14.2 | 8.13 |
| $(4, 5, 6)$ | 7.90 | 2.33 | 32.5 | - |

Table 3: **Partition ablation.**

matching task. On the other hand, as we increase $l$, accurate matching is promoted, as shown in the performance gaps among $(1, 0, 0), (1, 1, 1)$ and $(1, 1, 1, 1)$.

**Effects of varying $k$.** In Table 3, we summarize the effects of varying $k$, a hyperparameter used for dense zoom-in at inference. HPatches (Balntas et al., 2017) is used in measuring memory and run time. We take the maximum GPU memory utilization and average the run time. From the experiments, we find that varying $k$ has minor effects on performance while it has a large influence on memory and run time. This means that input images having resolutions similar to HPatches (Balntas et al., 2017) and ETH3D (Schops et al., 2017) do not require higher $k$ that leads to unnecessarily increased memory and run time; rather, smaller $k$ should be chosen. Note that to measure the maximum GPU utilization at $k = (4, 5, 6)$, we use 80GB A100 as RTX-3090 lacks GPU capacity for this configuration. Because of this, we omit the run-time, as it is unfair to compare with other configurations. For semantic matching, we empirically find that varying $k$ barely has an impact on the performance, while increasing the complexity. This is likely due to the relatively small resolutions of the image pairs in the standard benchmarks (Min et al., 2019b; Ham et al., 2016; 2017).

# E DISCUSSIONS

**Sparse and Quasi-Dense Evaluation.** In this section, we clarify the difference between the evaluation procedures adopted by COTR (Jiang et al., 2021) and its follow-up works (Revaud et al., 2022; Tan et al., 2022) to those of other existing works (Hong & Kim, 2021; Truong et al., 2020b;a; 2021b; Melekhov et al., 2019; Shen et al., 2020). COTR (Jiang et al., 2021) is one of the first works to use Transformer to find correspondences between images. Its follow-up works, including ECO-TR (Tan et al., 2022) and PUMP (Revaud et al., 2022), extend the work by improving performance or speed, or by reducing computations. These works attained state-of-the-art performance that significantly surpass the previous works, highlighting the effectiveness of these methods.

However, in order to compare with existing dense matching networks on dense correspondence datasets (Balntas et al., 2017; Schops et al., 2017), they first find sparse correspondences based on the confidence scores, where the correspondences below certain thresholds are discarded as mentioned in COTR (Jiang et al., 2021). Using only the sparse correspondences, AEPE or PCK is computed and compared to other works, which means that most of the erroneous correspondences that can significantly affect the metrics are not taken into account. On top of this, the densification is performed using only the confident sparse correspondences, and this version is indicated as "+interp" or dense version in their papers. The metrics are then calculated using only the points within the convex hull, and the points outside of the interpolated regions are discarded in the evaluation.

We find that this differs from conventional dense evaluation. Instead, this is more to be classified as "semi-dense" or "quasi-dense", which should be evaluated separately from existing dense methods. Although PUMP (Revaud et al., 2022) presents an extrapolation method to produce fully dense correspondence in their official implementation, the results reported in the paper appear to be computed using the confident quasi-dense flow maps, which clearly give it an apparent advantage similar to COTR (Jiang et al., 2021) and ECO-TR (Tan et al., 2022). Qualitative examples are shown in Fig. 3. We find that in the evaluation, COTR and its follow-up works have advantages over other existing dense matching works. Although COTR (Jiang et al., 2021) explicitly mentions in a footnote that their performance cannot directly be compared to existing dense methods, the presentation in the paper may lead to some confusion that complicates comparisons, especially when PUMP (Revaud et al., 2022) and ECO-TR (Tan et al., 2022) do not address this. Therefore, we provide these clarifications to reduce further potential confusion.

## F  PYTORCH-LIKE PSEUDO-CODE

We present Pytorch-like pseudo-code of the proposed method in Alg. **??**. In addition, pseudo-codes of the variants to show different aggregation strategies are presented in Alg. **??** to Alg. **??**.

## G  QUALITATIVE RESULTS

We provide more qualitative results on HPatches (Balntas et al., 2017) in Fig. 4, ETH3D (Schops et al., 2017) in Fig. 5 and Fig. 6 and SPair-71k (Min et al., 2019b) in Fig. 7 and Fig. 8. We also present more visualizations of PCA and the attention maps in Fig. 9 and Fig. 12.

## H  FUTURE WORKS

In this work, we explored distinctive characteristics of both feature and cost aggregations with Transformers. From the findings, we proposed a simple yet effective architecture that benefits from their synergy. However, more advanced techniques can be incorporated to further boost the performance. For example, as future work, we believe incorporating local-correlation maps to represent higher resolution cost volume would further improve the efficiency and performance improvements can be expected if $l$ is allowed to be increased given cheaper costs to represent cost volumes. However, a simple replacement of all the global correlations within the current architecture with that of local inevitably risks losing some information, which may degrade the performance. This means that a careful design would be necessary to achieve both high efficiency and performance. Another interesting extension is that as our model currently does not explicitly model matchability or uncertainty, it may have some disadvantages when handling occlusions. To compensate, we could design to output pixel-wise matchability scores and incorporate them into our framework, which we leave as future work.

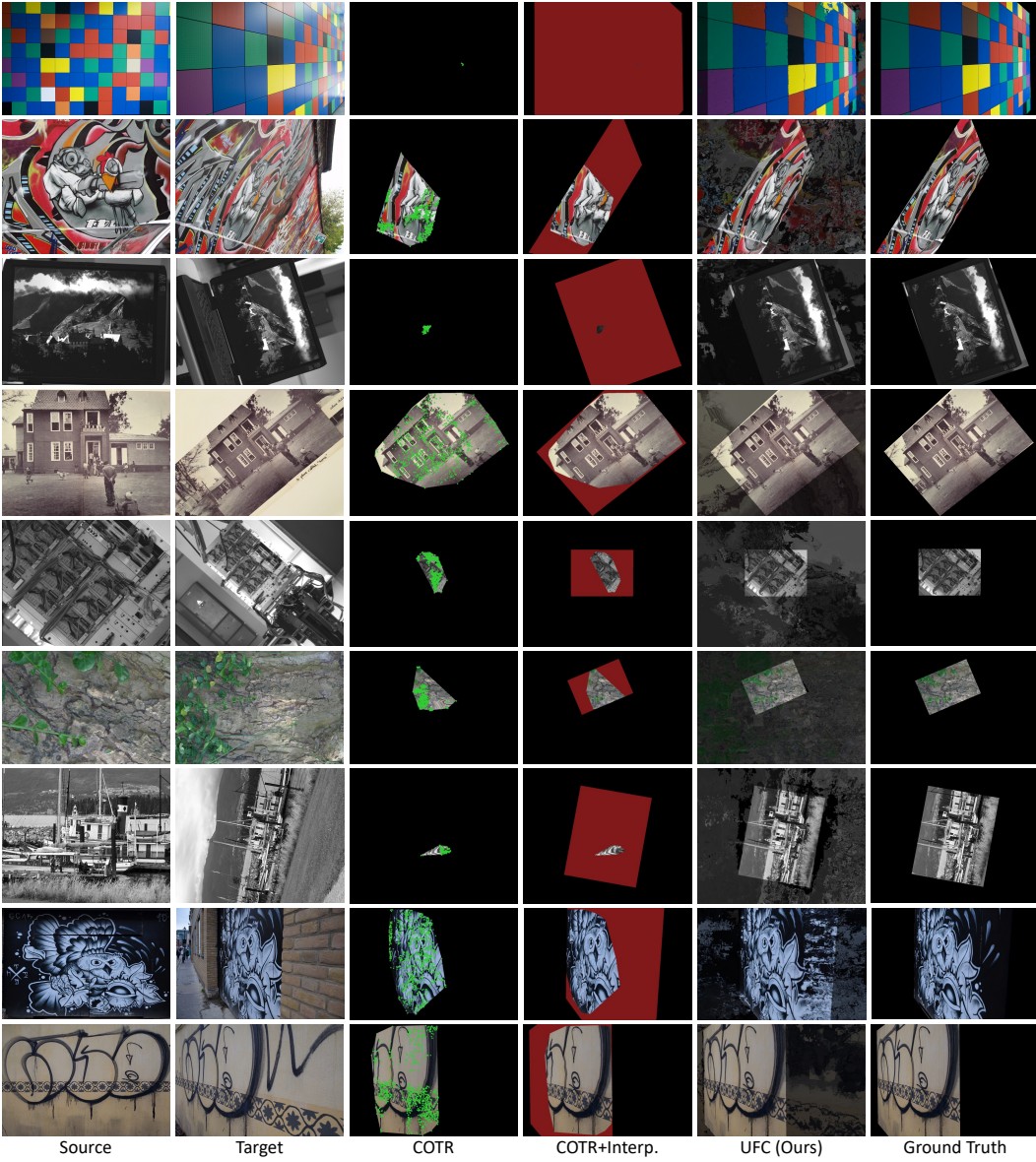

Figure 3: **Sparse, quasi-dense and dense comparisons.** For the sparse correspondence setting (COTR), only the confident correspondences, which are indicated as green dots, are used for evaluation. For quasi-dense correspondences (COTR+interp), the red regions are discarded at evaluation. Note that we visualize the warped image using the quasi-dense flow by COTR+interp, and then indicate the sparse points and discarded regions as green dots and red regions, respectively, on top of the warped images. For dense evaluation (UFC), every pixel correspondence is used for evaluation, and we highlight with a white box that our method successfully finds accurate dense correspondences.

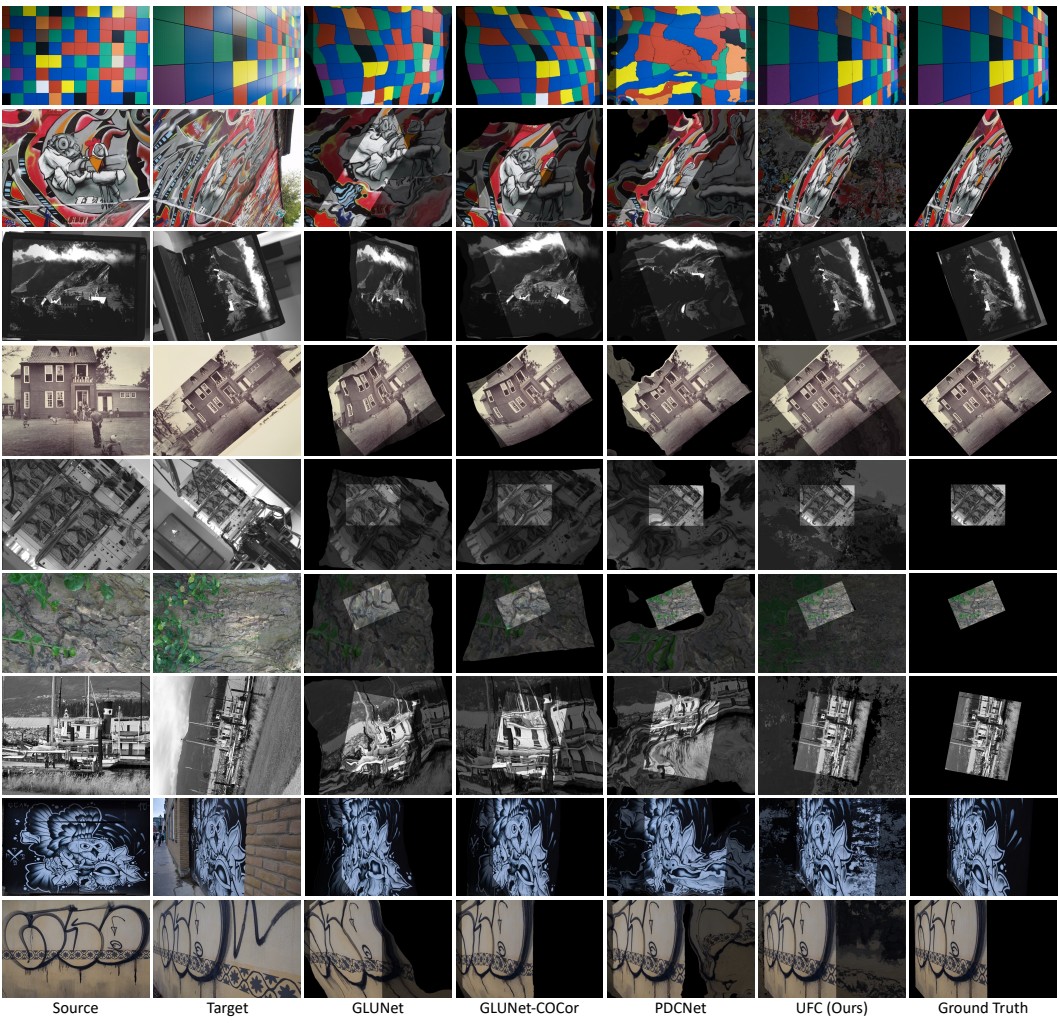

Figure 4: **Qualitative results on HPatches (Balntas et al., 2017).**

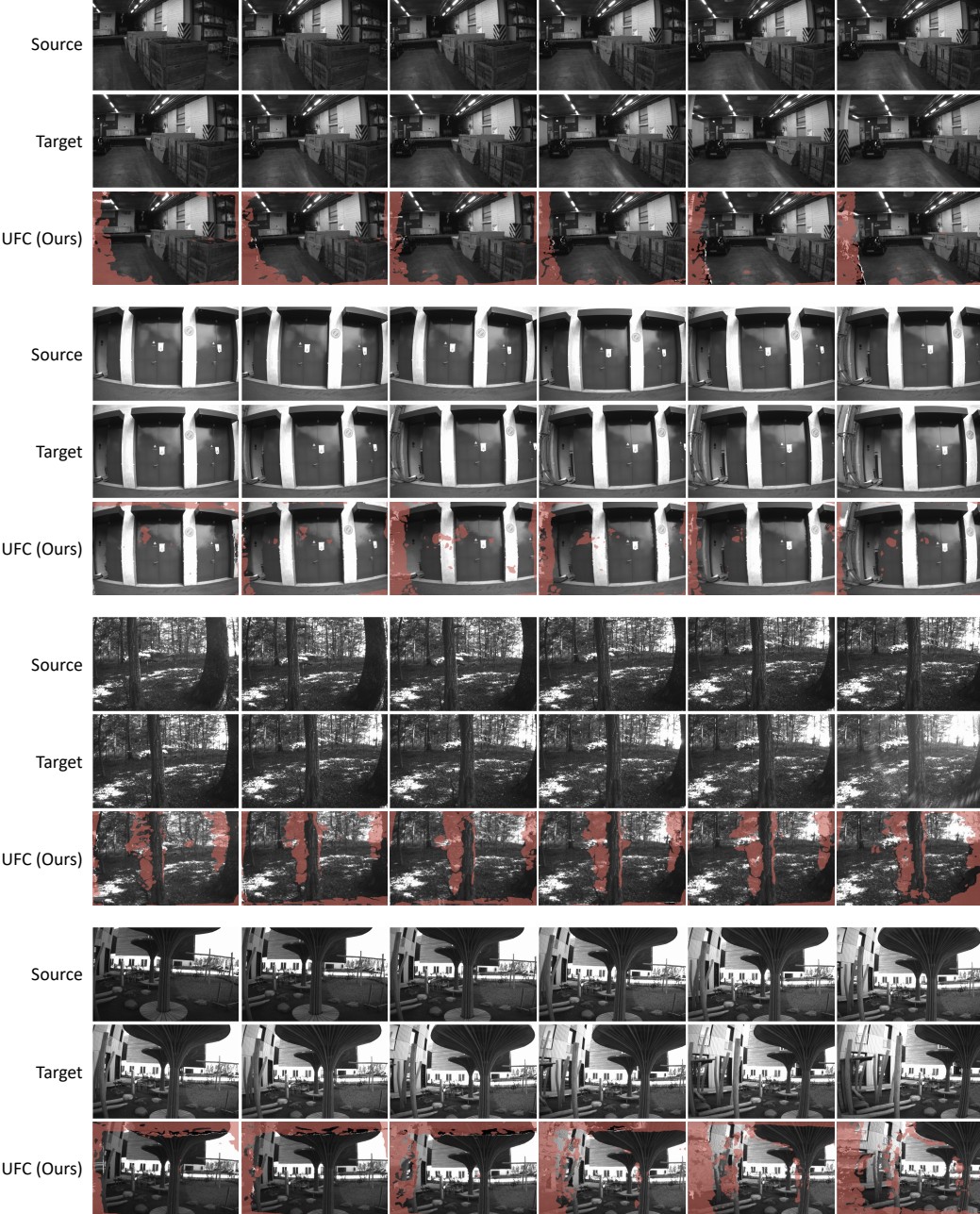

Figure 5: **Qualitative results on ETH3D (Schops et al., 2017).**

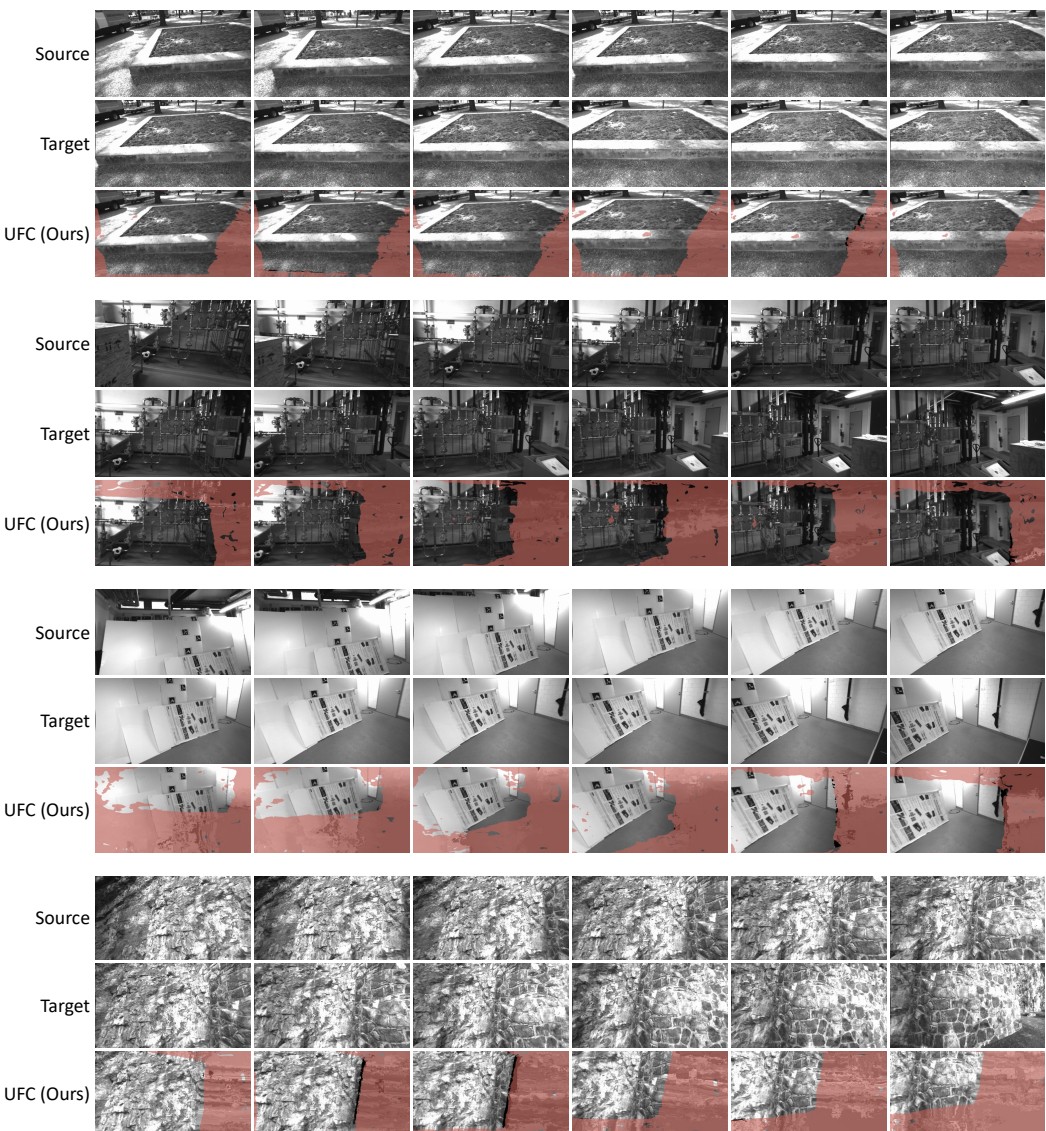

Figure 6: **Qualitative results on ETH3D (Schops et al., 2017).**

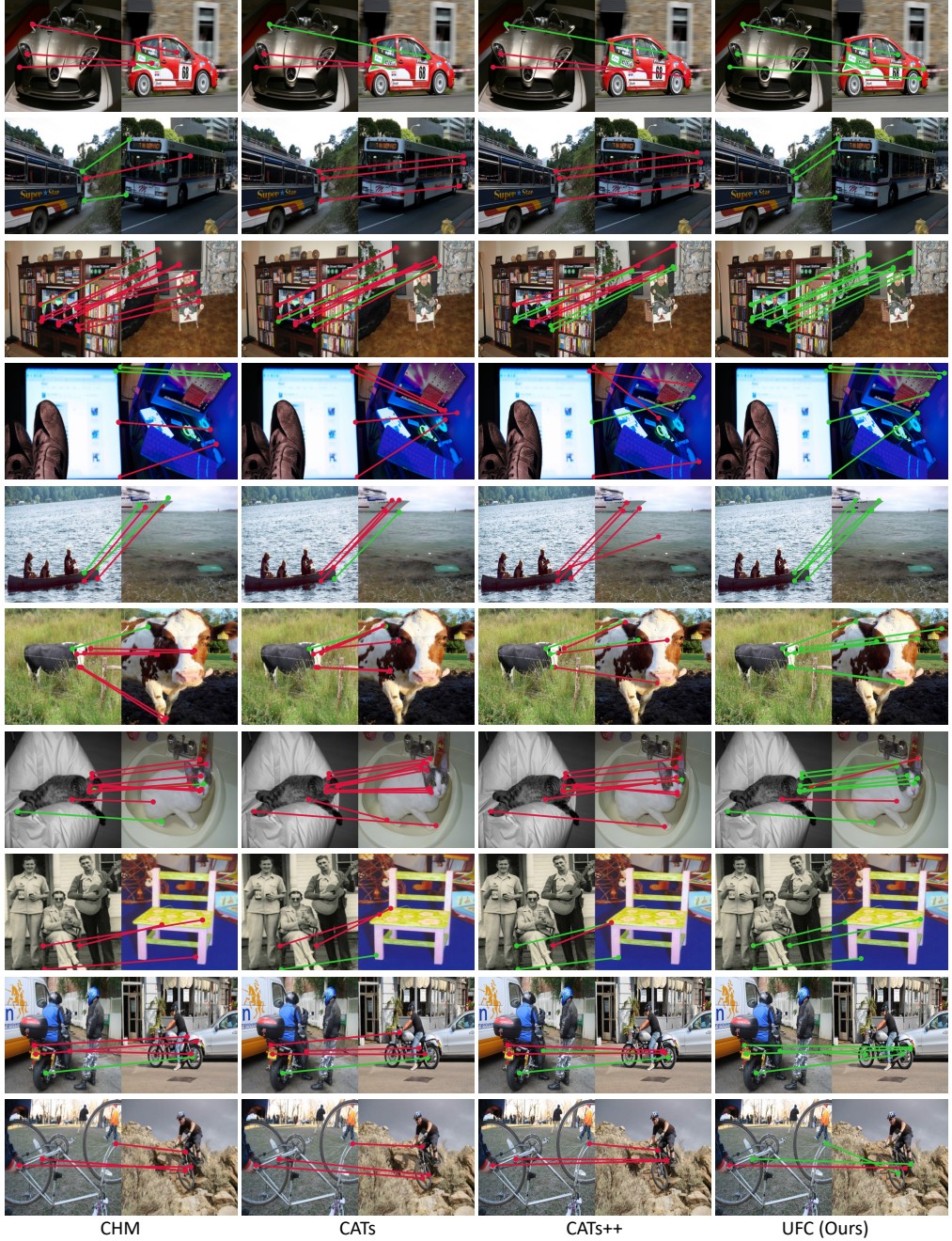

Figure 7: **Qualitative results on SPair-71k (Min et al., 2019b):** keypoints transfer results by CHM (Min et al., 2021), CATs (Cho et al., 2021), CATs++ (Cho et al., 2022a) and ours. Note that green and red lines denote correct and wrong predictions, respectively, with respect to the ground-truth.

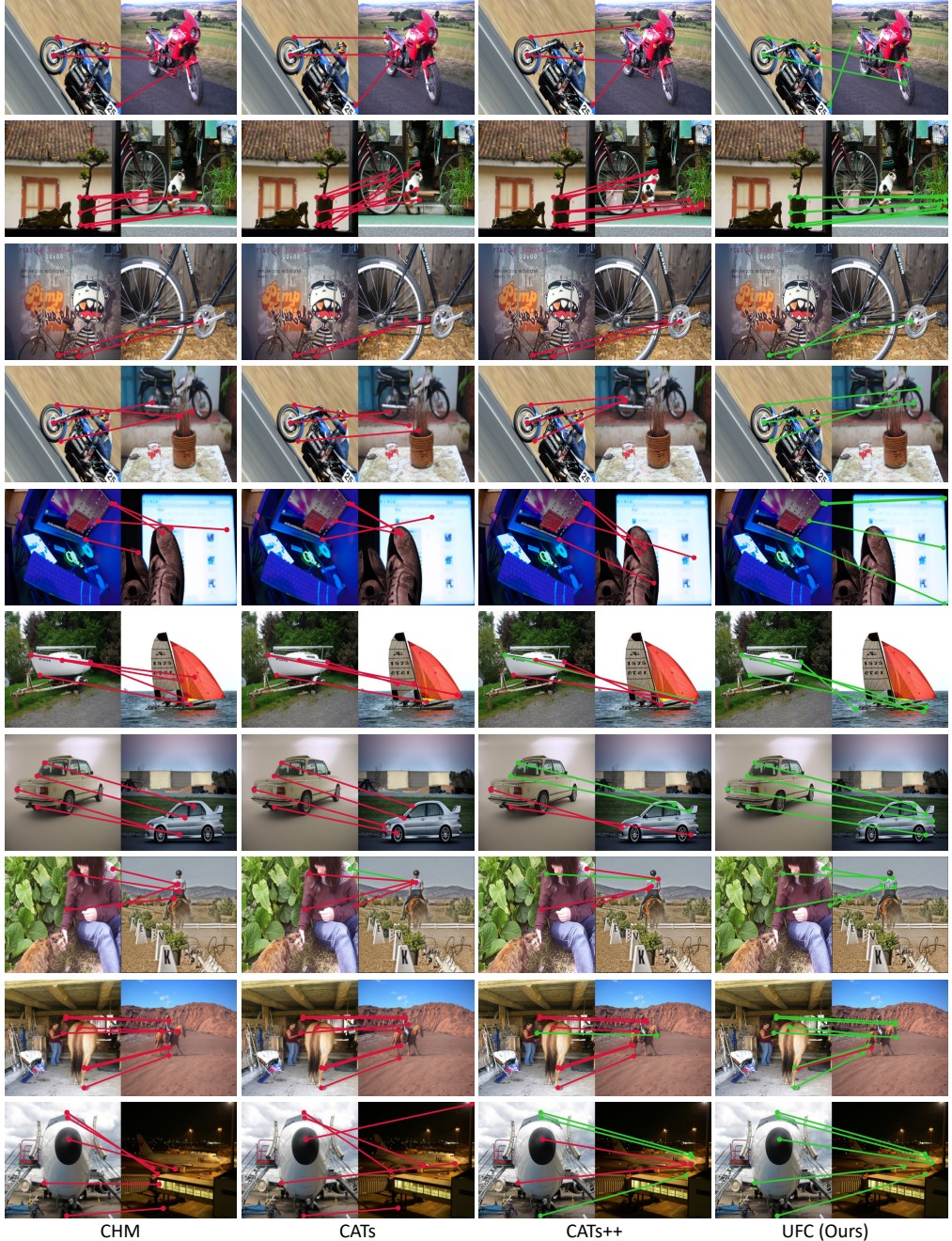

Figure 8: **Qualitative results on SPair-71k (Min et al., 2019b):** keypoints transfer results by CHM (Min et al., 2021), CATs (Cho et al., 2021), CATs++ (Cho et al., 2022a) and ours.

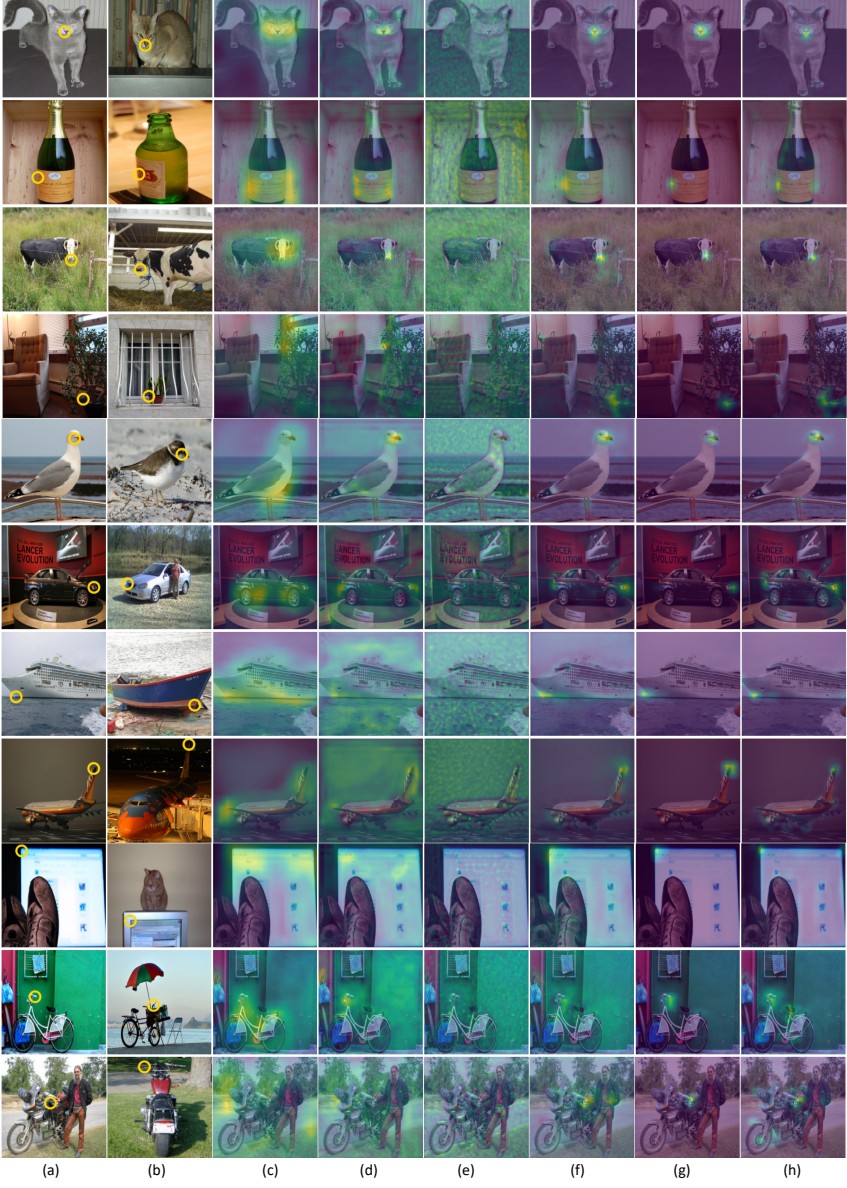

(a)    (b)    (c)    (d)    (e)    (f)    (g)    (h)

Figure 9: **Visualization of attention maps.**

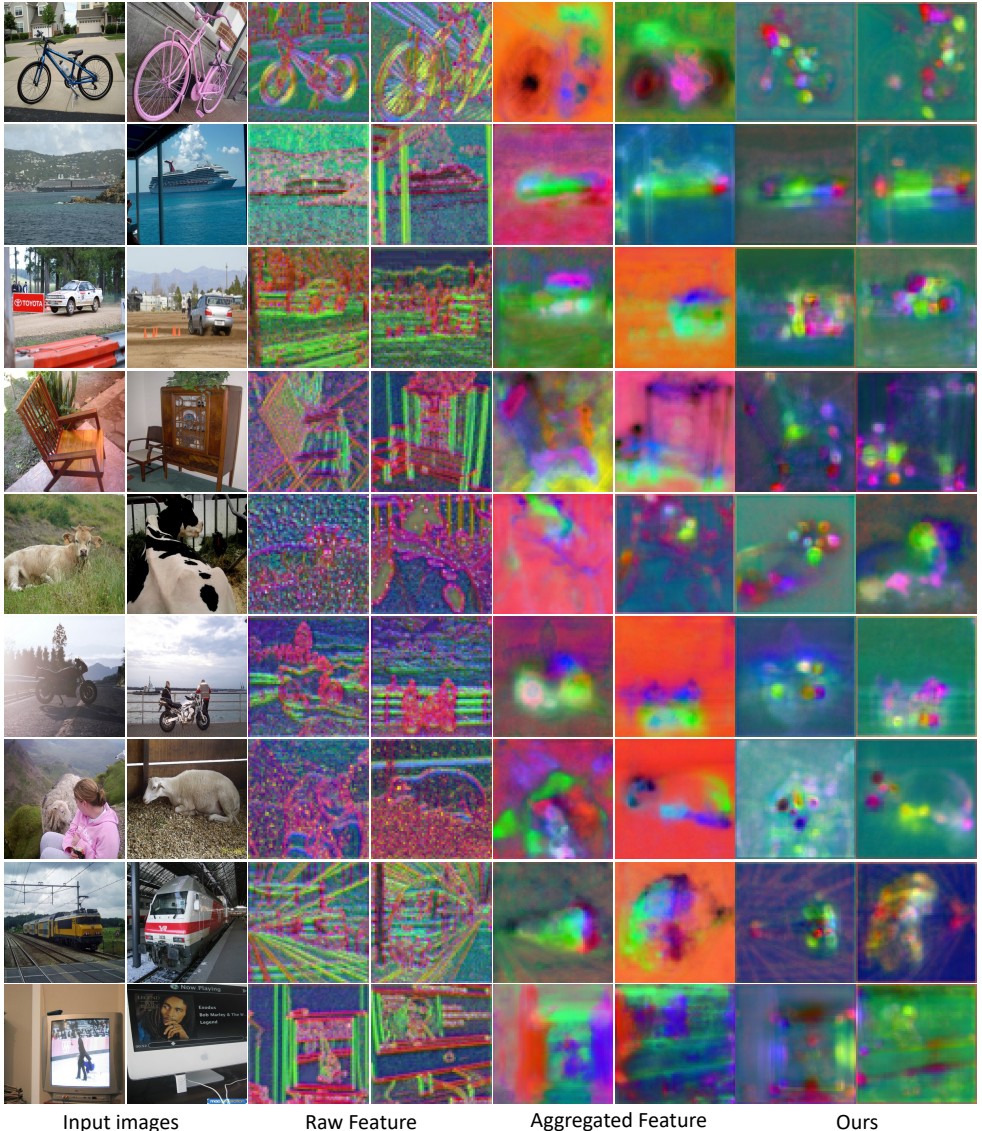

Figure 10: **Visualization of PCA results.**

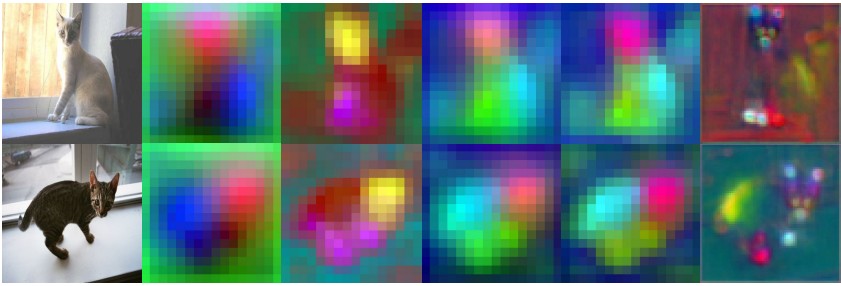

Figure 11: **Qualitative comparison of ablation studies in PCA Visualizaitons.** From left to right, the PCA visualizations of input images and those of (I-V) in the component ablation table are shown.

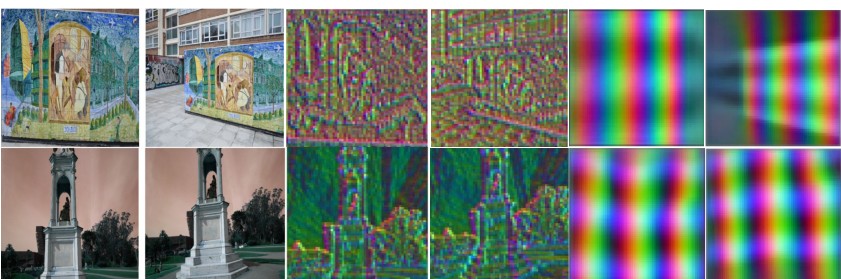

Figure 12: **PCA visualizations for geometric matching.** From left to right, source, target, raw features and the features after aggregations.