# OpenReview forum: "Unifying Feature and Cost Aggregation with Transformers for Semantic and Visual Correspondence"
_ICLR.cc/2024/Conference — ICLR 2024 poster_

### Official Review · Reviewer_HY22 · 2023-10-28

**Soundness:** 2 fair
**Presentation:** 2 fair
**Contribution:** 3 good
**Rating:** 5
**Confidence:** 4

**Summary:**

This paper proposed a dense matching estimation method by unifying both feature and cost volume aggregation with transformer. The authors first analyze the merits and faults of feature and cost aggregation and then claim that using them interleavely could improve the feature representation. Experiment results show the effectiveness of the proposed method.

**Strengths:**

1. To the best of my knowledge, this is the first work that discusses the relationship between feature and cost aggregation (learning). The authors carefully discuss their merits and faults in Sec.1 and Sec.3.
2. Interleavely aggregating features between feature and cost volumes is interesting. The visualization results of Fig.2, and Fig.3 also verify the effectiveness.
3. The authors provide complete experimental details in the supplementary.

**Weaknesses:**

1. The usage of "aggregation" is a little confusing. In my opinion, "aggregation" means combining multiple features into a single one, which should usually be used to describe the process of attention aggregation of ($QK^T$ and $V$). However, I am not sure whether "aggregation" is suitable to be used to indicate the whole learning process of cost volume learning. Because many cost volume learning is not related to attention learning.
2. Although the authors analyze the merits of feature/cost aggregation, some claims have not been clarified. For example, feature matching is "challenged by repetitive patterns and background clutters", while cost volume learning enjoys "robustness to repetitive patterns and background clutter". No evidence is shown in this paper to support this claim.
3. The authors did not formulate the method presentation well in Sec.4 and Fig.4, which makes the proposed method suffer from too complicated designs and difficult to follow. I strongly recommend the authors introduce the shape and reshape of the most important tensors to make the whole pipeline clearer. The concatenation in Eq(3) is operated along which dimension? Why $C'$ appears again in Eq.5 as $QK^T$, while $C'$ should be already defined as the output of the cost volume feature?
4. The experiments are not solid enough. The proposed method needs to be compared with more recent methods. In the geometric matching results from Tab2, most competitors are from 2020 and 2021, which are far from "state-of-the-art". Only one flow estimation method is considered (GMFlow). However, as discussed in the supplementary, the comparison is not fair, because GMFlow is trained on Sintel rather than DPED-CityScape-ADE+MegaDepth fine-tuning. Besides, all these competitors are trained on DPED-CityScape-ADE **or** MegaDepth (the proposed method is trained with DPED-CityScape-ADE **and** MegaDepth finetuning) as said in supplementary B.1. The authors should clarify this.

**Questions:**

As discussed in the related works, many stereo-matching and optical flow works use Transformer-based cost aggregation networks.
The idea proposed in this paper should be a general way to improve all feature matching-based tasks, and I think there are no enormous model differences among stereo, flow, and dense matching. So the authors should compare these SOTA stereo and flow methods in a more fair way. For example, re-training the model with the same data setting for dense matching or verifying the effectiveness of the proposed method in stereo and flow estimation benchmarks.

---

> ### Author Response · Authors · 2023-11-16
> **Response to the reviewer HY22 (1/2)**
>
> We thank the reviewer for the constructive comments! Our responses can be found below:
>
> > Unsure whether the term "aggregation" is suitable for operations performed over cost volumes.
>
> In the stereo matching literature, "cost aggregation" plays a pivotal role. This process involves combining primary matching costs, typically through a filtering process, to generate a comprehensive disparity map, as noted in source [A]. The scope of "aggregation" in this context is broad and versatile. It can encompass various techniques ranging from the application of simple Gaussian kernels to more complex methods like 3D convolutions, as highlighted in source [B]. Additionally, the term also covers techniques such as semi-global or local attentions, as well as other forms of attention mechanisms, indicated in source [C].
>
> These diverse aggregation techniques serve to illustrate the multifaceted nature of cost volume learning in stereo matching. The term "aggregation" is not merely confined to the domain of attention-based feature combination but extends to a variety of methods, each contributing to the accurate and efficient processing of stereo imagery. This wide-ranging application of the term confirms its appropriateness and relevance in the field, demonstrating that cost volume learning indeed benefits from and incorporates a spectrum of aggregation strategies. However, we can change the term to "refinement" if it better fits to describe our approach.
> We highly appreciate the reviewer's comment in this regard.
>
> [A] Cross-Scale Cost Aggregation for Stereo Matching
> [B] Correlate-and-Excite: Real-Time Stereo Matching via Guided Cost Volume Excitation
> [C] GA-Net: Guided Aggregation Net for End-to-end Stereo Matching
>
> > Evidences to claim that feature matching is challenged by repetitive patterns and background clutters and cost aggregation enjoys robustness to repetitive patterns and background clutter.
>
>
> There is a well-established understanding in computer vision that feature matching can be particularly challenging in scenes with repetitive patterns and cluttered backgrounds. This difficulty arises from the reliance of feature matching techniques on identifying and aligning distinct keypoints or features between images. In environments where textures are repetitive or backgrounds are complex, these features often become ambiguous, leading to mismatches and inaccuracies. This limitation is widely recognized in the literature and has led to the development of enhanced feature detection algorithms, such as SIFT, and more recent methods like LOFTR, which aim to refine local features to extract high-quality matches in indistinctive or low-texture regions.
>
> To support our claim, we will reference these matching methods in our paper. They exemplify the ongoing efforts in the field to address the challenges posed by such complex environments. However, we acknowledge that the assertion regarding the robustness of cost aggregation to these challenges might be contentious. Nevertheless, while our paper does not provide specific empirical evidence to support this claim (although we provide visualizations that can explain such robustness in Fig. 4 to some extent), it is nonetheless informed by a substantial body of research in the semantic matching literature. Numerous studies in this area have demonstrated the effectiveness of cost aggregation in various contexts.
>
> Nevertheless, in light of the comment, we agree that it is prudent to moderate our claim regarding the robustness of cost aggregation to environments with repetitive patterns and cluttered backgrounds. We recognize that this aspect could be seen as a hypothesis rather than a conclusively proven fact. We appreciate the constructive comment and will accordingly adjust the tone of our claim to reflect a more cautious stance on this matter.

---

> > ### Comment · Reviewer_HY22 · 2023-11-21
> > **Thanks for the response from the authors and additional experiments**
> >
> > Thanks for the response from the authors and additional experiments.
> >
> > 1. The concern about "aggregation" is addressed by the additional illustration from the authors. But I still recommend to provide some discussion about this term in the paper.
> >
> > 2. For the claim about solving "challenged by repetitive patterns and background clutters", the authors explain the importance of this in feature matching. I am already aware that tackling this issue is an intricate and widely recognized problem within the domain of feature matching. Thus the way to handle it attaches much attention to this field. Regrettably, the authors commit to tuning down the claim, diminishing the overall contribution of this paper.
> >
> > 3. For the experiment results compared to DKM, I think the authors should further provide the results of AUC 3/5/10px as shown in the DKM paper to make the comparison more solid.
> >
> > 4. For the results of flow estimation, the improvement is not very significant compared to PDC-Net+ (KITTI) and GMFlow (SINTEL).

---

> ### Author Response · Authors · 2023-11-16
> **Response to the reviewer HY22 (2/2)**
>
> > The dimension of concatenation in Eq.3.
>
>
> We apologize for any confusion caused by the presentation of our methodology and appreciate the opportunity to clarify. To enhance understanding, we will include the dimensions of all tensors in our method section.
>
> Addressing the reviewer's query regarding the concatenation process, it is performed as follows: We start with a cost volume of shape h×w×h×w. To this, we concatenate a feature map of shape h×w×c, treating one of the h×wh×w dimensions as the spatial dimension. This results in a combined shape of h×w×(hw+c), effectively merging the cost volume and feature map information.
>
> Regarding Equation 5, the use of C′ is an intentional workaround to avoid the conventional computation of the attention map, typically represented as QKT in standard attention mechanisms. Instead of computing the attention map traditionally, we utilize the cost volume to transform it into a matching distribution. This approach effectively functions as a cross-attention map. The effectiveness of this method is validated in Table 4, where we demonstrate its practical utility and performance. We hope this explanation clears up any ambiguities.  If there are further questions or additional clarifications needed, we are more than willing to provide further details.
>
> > Include more recent works in Table 2, and GMFlow is trained on different dataset. Verify effectiveness of the proposed method in optical flow.
>
> We apologize for any oversight in our initial presentation of recent methods in our comparative analysis. While Table 1 comprehensively includes recent methods, we acknowledge that Table 2 appeared somewhat outdated. To address this, we have now included an evaluation of DKM, a method that has been tested on HPatches, with its results available in its public GitHub repository. We have compared these results with those of our method:
>
> | Methods           |   I  |   II |   III | IV    | V     | Avg   |
> |-------------------|:----:|-----:|------:|-------|-------|-------|
> | PDC-Net (CVPR'21) | 1.15 | 7.43 | 11.64 | 25.00 | 30.49 | 15.14 |
> | DKM (CVPR'23)     | 2.35 | 7.24 | 16.48 | 29.86 | 39.35 | 19.06 |
> | Ours              | 1.91 | 6.13 | 5.62  | 6.36  | 19.44 | 7.88  |
>
> As illustrated, our method demonstrates highly competitive performance, even when juxtaposed with newer methods like DKM.
>
> Regarding the comparison with GMFlow, we recognize and apologize for the previously unfair comparison. Unlike other methods in our comparison that were trained on MegaDepth, GMFlow was not, which might have skewed the results.  To address this and to further evaluate the generalizability of our method as suggested by the reviewer, we conducted additional evaluations in the context of optical flow. For these evaluations, we utilized the KITTI and SINTEL datasets. Aligning with the practices of GLU-Net, PDC-Net, and GoCOR, our method was not trained on the Chair and Things datasets but rather on MegaDepth. The results of these additional evaluations are presented below:
>
> | Methods  | KITTI-2015 | SINTEL (Final) |
> |----------|:----------:|-------:|
> | RAFT     |    5.32    |   2.69 |
> | PDC-Net  | 5.40       | 4.54   |
> | PDC-Net+ |    4.53    |      - |
> | GMFlow   | 7.77       | 3.44   |
> | Ours     | 4.39       | 4.16   |
>
> From the results, while ours outperforms others in KITTI-2015, it is observed that ours achieves lower AEPE than existing flow methods. However, it should be noted that when compared to PDC-Net and PDC-Net+, ours consistently performs better, which validates the effectiveness of our approach in optical flow task as well. We acknowledge that newer, more sophisticated technical designs may emerge in the future, but we wish to emphasize that while numerous architectural designs, e.g., convex upsampling, uncertainty module or GOCor-like correlation optimization scheme, could potentially be included and enhance the synergy between feature aggregation and cost aggregation, our framework works as a stepping stone that stands out for its simplicity and effectiveness.

---

> > ### Comment · Area_Chair_pK98 · 2023-11-21
> > **Dear reviewer**
> >
> > Please try your best to engage during the discussion period.

---

> ### Author Response · Authors · 2023-11-21
> **Thanks for the reply!**
>
> Thanks for the reply!
>
> 1. We wish to emphasize that although we said that we will tone down the sentence, this does not mean that the substantial body of study in the semantic matching literature has been adopting cost aggregation for no benefits. It has been recognized by several works that cost aggregation enjoys strong generalization power [A, B], many works in semantic matching adopts cost aggregation to account for the background clutters, repetitive patterns and intra-class variations and finally, we also clearly showed adopting cost aggregation boosts the performance. Moreover, our contribution should not be diminished from the toned down claim, but rather be enhanced since we managed to show  the improved performance in semantic matching. This is an evidence that our method better considered such challenges.
>
> 2. We agree that for solid comparisons, we should add DKM if we include pose estimation and visual localization. However, we wish to highlight that these sparse matching tasks are beyond our paper's scope and  we **already evaluated our method on 5 dense correspondence benchmarks, and also yielded competitive performance on additional benchmarks we used for this rebuttal.** We also discussed a limitation that the typical dense matching method faces for sparse matching tasks (while DKM is also a dense method and it attains SOTA, it's their main contribution that they propose a method that overcomes this drawback. Our contribution lies more on thorough and comprehensive investigation and focus on dense tasks rather than achieving SOTA on every matching related tasks). We also outperforms DKM in HPatches, which we believe this indicates some trade-offs. Similar to any other works, we believe that it would not be fair to penalize our work for not including as many as 10 datasets (if we include sparse matching tasks) from different tasks.
>
> For the small improvements in Optical flow datasets, we wish to stress that not only it should not diminish our contribution because we don't outperform them by significantly large margin, but also it should be noted that it is quite common for most (even the very recent papers) optical flow methods to also report similar extent of improvements every year (We respectfully refer the reviewer to paperswithcode. The changes in EPEs are mostly decimal points). The focus should be on how we obtain better results than PDCNet+, as we adopt similar training strategy and exceeding their performance should sufficiently demonstrate the effectiveness of the unified aggregation approach we introduce.
>
> We hope that the merits of our approach is recognized, as we have shown that our approach achieved highly strong performance in 5 different benchmarks (with large margin.). We sincerely hope that the contributions of  our work is not ignored for not being able to achieve SOTA in tasks beyond the paper's scope, since MegaDepth, ScanNet and YFCC100M are used for evaluating sparse matching tasks in outdoor and indoor pose estimation, respectively, while our focus is on dense matching.
>
>
>
>
> [A] A simple and efficient approach for adaptive stereo matching.
> [B] Graftnet: Towards domain generalized stereo matching with a broad-spectrum and task-oriented feature.

---

> > ### Author Response · Authors · 2023-11-22
> > **Additional comparison**
> >
> > We provide an additional comparison to clarify our toned down claim.
> >
> > To further validate the effectiveness of cost aggregation and our approach in semantic matching, where background clutters, repetitive patterns and intra-class variations pose additional challenges, we directly compare with GLU-Net architecture that shares similar architecture to PDC-Net and PDC-Net+. They are trained on PF-PASCAL and evaluated on both PF-PASCAL and PF-WILLOW, following the standard protocol.
> >
> > | Model   | PF-PASCAL  |            | PF-WILLOW  |            |
> > |---------|------------|------------|------------|------------|
> > |         | alpha=0.05 | alpha=0.1  | alpha=0.05 | alpha=0.1  |
> > | GLU-Net | 48.4       |    72.4    | 39.7       | 67.6       |
> > | Ours    | 88.0       | 94.8       | 50.4       | 74.2       |
> >
> > The table above clearly differentiates the performance, which validates the effectiveness of our approach in this challenging task. Since tomorrow is the deadline for the rebuttal period, should you have any further questions, please don’t hesitate to let us know. Thank you for your time and effort.

---

### Official Review · Reviewer_gxEA · 2023-10-30

**Soundness:** 2 fair
**Presentation:** 3 good
**Contribution:** 2 fair
**Rating:** 5
**Confidence:** 4

**Summary:**

This paper proposes to combine feature and cost aggregation to address the dense feature matching task.

The main idea is to use cost score matrix for both self- and cross-attention feature updating, to learn more discriminative features and compute better cost matrix.

Experiments on some semantic and geometric matching datasets show the effectiveness of the proposed method.

**Strengths:**

1) This paper is generally presented well;

2) The idea of use cost score matrix for both self- and cross-attention feature updating is simple and effective;

3) Experimental results are good.

**Weaknesses:**

1)  Section 5.1. It is quite blurry for me whether previous state-of-the-art methods in Table 1 and 2 are trained on the same datasets; For example, the proposed method is trained on the DPED-CityScape-ADE and MegaDepth datasets;

2) The performance of the proposed method on the optical flow (KITTI, Sintel) task is blurry for me;

3) It's good to see the improved matching performance on the HPatches dataset. However, I want to see whether the improved matching would lead to better Rotation and translation estimations.

4) Using cost score matrix for both self- and cross-attention feature updating is good. However, this contribution may be constrained to large overlapping ratio between images. If pairwise images have small overlapping ratio, the cost score matrix is noisy, and may provide wrong guidance for feature updating. Would you please check whether the proposed method works on some challenging image pairs from the MegaDepth dataset.

5) Please show some failure cases.

**Questions:**

Please refer to the weaknesses.

---

> ### Author Response · Authors · 2023-11-16
> **Response to the reviewer gxEA (1/2)**
>
> We thank the reviewer for the thorough reviews. Our response can be found below :
>
> > Whether Table 1 and 2 competitors are trained on the same datasets:
>
> In Table 1, to ensure a fair and consistent comparison, all methods were trained and evaluated on the same datasets. Specifically, for evaluations on SPair-71k, all methods were trained on SPair-71k, and for evaluations on PF-PASCAL and PF-WILLOW, they were trained on PF-PASCAL. This uniform approach guarantees that the results in Table 1 are directly comparable, as each method was subjected to the same training and testing conditions.
>
> For Table 2, we acknowledge that there was some variation in the training strategies employed. All methods, with the exception of GMFlow, were trained either on MegaDepth or a combination of DPED, ADE20K, and Cityscapes (DPED-ADE-Cityscape). Notably, except for DMP, which is trained on DPED-ADE-Cityscape, all other methods were trained on MegaDepth. While this introduces some variation, the majority of methods being trained on MegaDepth still allows for a reasonably fair comparison. However, we recognize and apologize for the potential unfairness in the comparison with GMFlow, which was not trained on MegaDepth.
>
> To address this and to further evaluate the generalizability of our method as suggested by the reviewer, we conducted additional evaluations in the context of optical flow. For these evaluations, we utilized the KITTI and SINTEL datasets. Aligning with the practices of GLU-Net, PDC-Net, and GoCOR, our method was not trained on the Chair and Things datasets but rather on MegaDepth. The results of these additional evaluations are presented below:
>
> | Methods  | KITTI-2015 | SINTEL (Final) |
> |----------|:----------:|-------:|
> | RAFT     |    5.32    |   2.69 |
> | PDC-Net  | 5.40       | 4.54   |
> | PDC-Net+ |    4.53    |      - |
> | GMFlow   | 7.77       | 3.44   |
> | Ours     | 4.39       | 4.16   |
>
> From the results, while ours outperforms others in KITTI-2015, it is observed that ours achieves lower AEPE than existing flow methods. However, it should be noted that when compared to PDC-Net and PDC-Net+, ours consistently performs better, which validates the effectiveness of our approach in optical flow task as well. We acknowledge that newer, more sophisticated technical designs may emerge in the future, but we wish to emphasize that while numerous architectural designs, e.g., convex upsampling, uncertainty module or GOCor-like correlation optimization scheme, could potentially be included and enhance the synergy between feature aggregation and cost aggregation, our framework works as a stepping stone that stands out for its simplicity and effectiveness.
>
> We hope this additional information and the results of our extended evaluations address the concerns raised and provide a more complete understanding of the capabilities of our method.
>
> > Optical flow transferability.
>
> We provide quantitative results above.

---

> ### Author Response · Authors · 2023-11-16
> **Response to the reviewer gxEA (2/2)**
>
> > Pose estimation evaluation
>
> Following reviewer's suggestion, we perform additional evaluation on pose estimation. Specifically, we use YFCC100M and ScanNet. The results are shown below:
>
> | Methods            | 5$\degree$ | 10$\degree$ | 15$\degree$ |
> |--------------------|:----------:|------------:|------------:|
> | SP+SG (CVPR'19)    |    38.7    |   59.13 | 75.8        |
> | LOFTR (CVPR'21)    |    42.4    |        62.5 | 77.3        |
> | PDCNet+ (TPAMI'23) |    35.51   |       58.08 | 74.50       |
> | Ours               | 37.62      | 59.83       | 73.22       |
>
> | Methods            | 5$\degree$ | 10$\degree$ | 15$\degree$ |
> |--------------------|:----------:|------------:|------------:|
> | SP+SG (CVPR'19)    |    16.16   |       33.81 | 51.84       |
> | LOFTR (CVPR'21)    |    16.88   |   62.533.62 | 77.350.62   |
> | 3DG-STFM (ECCV'22) | 23.6       | 43.6        | 61.2        |
> | PDCNet+ (TPAMI'23) |    20.25   |       39.37 | 57.13       |
> | Ours               | 20.23      | 39.21       | 57.77       |
>
> The upper table is YFCC and the below is ScanNet.As shown, although it surpasses PDCNet and others, it is apparent that our method is not  state-of-the-art. Nevertheless, we wish to highlight that our formulation, which  aims at  dense matching, should be separated from sparse matching (feature matching) works. A pervasive challenge in dense matching methods is their requirement to extract all matches between views. This challenge often results in many dense matching networks facing difficulties in sparse or semi-sparse methods for geometry estimation,  which is shared by many other works. Our framework is also not immune to this limitation. A specific area where our current architecture falls short is in its approach to handling confidence scores. While our cost volumes do encode a confidence score for each tentative match, a recent study like Lightglue has shown the benefits of explicitly estimating these confidence scores directly from the network, particularly for sparse matching tasks. This explicit estimation technique enhances performance in challenging scenarios, such as those involving occlusions or uncertain regions. Through empirical experiments, we've found that our model struggles more in these situations, largely attributed to the absence of a dedicated confidence estimation module.   To provide a complete understanding of our framework's capabilities and limitations, we will introduce a new section in our paper dedicated to discussing this limitation.
>
> Moreover, we wish to stress that the true essence and value of our work lie in the comprehensive and thorough investigations we have conducted into the relationships and synergies between feature aggregation and cost aggregation, as appreciated by the reviewer HY22, especially in the context of dense matching tasks. Our research goes beyond merely presenting a new technical design for state-of-the-art performance. Instead, it offers a deeper understanding of the underlying principles and interactions within these aggregation methodologies. This in-depth exploration, we believe, provides a more substantial and lasting contribution to the field.
>
> > Whether the proposed method works on minimal overlapping image pairs (MegaDepth Evaluation)
>
> We agree that if the cost score matrix is noisy, it may provide wrong guidance for feature updating. However, as shown in qualitative results on semantic matching datasets in the supplementary material, our method performs quite well given extreme scale differences, and as there are numerous image pairs with extreme geometric scenarios as well as additional challenges like background clutters and intra-class variations, we may assume that our method is quite competitive. Nevertheless, following the reviewer's suggestion, we also try evaluating our method on the MegaDepth dataset and the results are summarized below:
>
>
> | Methods                    | 1px   | 3px   | 5px   |
> |----------------------------|-------|-------|-------|
> | PDC-Net (D) (CVPR 2021)    | 68.95 | 84.07 | 85.92 |
> | PDC-Net (MS) (CVPR 2021)   | 71.81 | 89.36 | 91.18 |
> | PDC-Net+ (H) (TPAMI 2023)  | 73.9  | 89.21 | 90.48 |
> | Ours                       | 73.54 | 91.71 | 92.99 |
>
>
> While ours achieves competitive performance, we agree that ours may struggle when image pairs with large geometric deformations are given. We will be sure to add the limitations and future directions of our method in the manuscript.
>
> > Failure cases
>
> We will visualize failure cases for both geometric and semantic matching and include the limitation discussed above as well. We appreciate the reviewer's comment.

---

> > ### Comment · Area_Chair_pK98 · 2023-11-21
> > **Dear reviewer**
> >
> > Please try your best to engage during the discussion period

---

### Official Review · Reviewer_4EyK · 2023-10-31

**Soundness:** 3 good
**Presentation:** 3 good
**Contribution:** 3 good
**Rating:** 6
**Confidence:** 3

**Summary:**

This paper presents a new vision transformer architecture to conduct feature aggregation and cost aggregation for dense matching tasks. The authors show distinct characteristics of feature aggregation and cost aggregation. and use self- and cross-attention mechanisms to unify the feature and cost aggregation. They validate the effectiveness of the proposed method with semantic matching and geometry matching.

**Strengths:**

1. The idea to unify feature aggregation and cost aggregation is interesting. It compensates for the lack of semantic information in cost representation and helps to drive the features in each image to become more compatible with others.
2. They conduct extensive experiments on semantic matching and geometry matching to validate the effectiveness of the proposed method UFC. UFC can improve the matching performance. And they provide step-by-step ablations of each component.
3. The paper is well-organized and easy to understand. The authors visualize the changes in feature maps and cost volumes, which helps understand how their method works. I see that feature aggregation can preserve semantic information and geometry structure and the cost aggregation reduces the noise in cost volumes.

**Weaknesses:**

1. In visualization results Figure 2, features with integrative aggregation methodology preserve the semantic information. However, it seems the proposed method damages the local discriminative ability of features.

**Questions:**

1. The local discriminative ability of features is also important for dense matching tasks. I would like to see an analysis of whether the proposed method causes damage in this perspective or whether this issue can be avoided in some design of the method.

---

> ### Author Response · Authors · 2023-11-16
> **Response to the reviewer 4EyK**
>
> We thank the reviewer for the reviews. Our response is shown below :
>
> > An analysis of the local discriminative ability of features.
>
> One of the goals of our proposed aggregation is also to synthesize and incorporate broader contextual information. This often involves balancing local feature details with more global, semantic information. While the process enhances the overall semantic understanding of the image, it can sometimes appear to diminish the acuity of local features. This is not necessarily a flaw but rather a trade-off, aiming for a more holistic understanding of the image data. As demonstrated in Fig. 4 of the supplementary material, top row, where image pairs have extreme geometric deformation and repetitive patterns, our framework can warp all pixels to the corresponding locations unlike others. This shows our framework does not necessarily experience such damage.

---

> > ### Comment · Area_Chair_pK98 · 2023-11-21
> > **Dear reviewer**
> >
> > Please try your best to engage during the discussion period

---

### Official Review · Reviewer_96VU · 2023-11-01

**Soundness:** 3 good
**Presentation:** 3 good
**Contribution:** 2 fair
**Rating:** 6
**Confidence:** 4

**Summary:**

The paper introduces an integrative feature and cost aggregation modules in a CNN architecture for a semantic correspondence task. The introduced module cleans up noisy matches in the cost volume and thus improves the matching accuracy. The paper demonstrates better accuracy on semantic matching and geometric matching tasks by using their method.

**Strengths:**

- Good results

  The paper achieves good accuracy on both semantic and geometric matching tasks (Table 1 and 2). It demonstrates the effectiveness of the proposed aggregation modules.

- Detail analysis

  The paper provides a sufficient amount of analysis. Fig. 3 visualizes the qualitative comparison of the proposed modules (from (f) to (h)). Further, the ablation study (Table 3 and 4) validates the proposed ideas.

**Weaknesses:**

Despite the good accuracy on both tasks, there are concerns about novelty/contributions.

- Existing ideas in other literature

  Similar ideas on feature and cost volume aggregation have been demonstrated in other literature such as stereo matching [a,b] and optical flow estimation [c]. Actually the related work section (Sec. 2) summarizes those relevant papers very well. Compared to the existing solutions, what would be the new technical design (in self-/cross-attention) of the proposed module, except for applying it to semantic & geometric matching problems? Can the newer technical design from the proposed module also benefit other tasks that use cost volume, eg., stereo matching, optical flow, scene flow, etc. ?

   [a] Attention-Aware Feature Aggregation for Real-time Stereo Matching on Edge Devices, ACCV 2020

   [b] Attention Concatenation Volume for Accurate and Efficient Stereo Matching, CVPR 2022

   [c] GMFlow: Learning Optical Flow via Global Matching, CVPR 2022

- Limitation

  Discussion on the limitation is missing. What would be the limitation of the method or unsolved problems?


- Can the paper provide more qualitative examples and discuss where the gain mainly originates?

  Table 4 shows the accuracy improvement by adding more components. I am wondering if the paper can also include some qualitative examples and discuss where the gain mainly originates, such as resolving some particular matching ambiguity. It would be great if the paper can provide more insights related to its improvement.

**Questions:**

- Increase of learnable parameters

  How many number of learnable parameters account for the new module (i.e., integrative feature and cost aggregation module)? How significant are they compared to the number of parameters of the entire network (15.5M)? Probably it's also good to include an extra column in Table 3 and 4 for the number of network parameters.

- Resolution of the cost volume

  What's the resolution of the cost volume (saying the input image resolution is HxW)? At each level the features are upsampled ($D^{l}_s$), but how can the resolution of the cost volume ($C^{l})$ remain the same over different pyramid levels? Is there any reason to fix the resolution of the cost volume?

---

> ### Author Response · Authors · 2023-11-16
> **Response to the reviewer 96VU (1/3)**
>
> We highly appreciate for the reviewer's constructive comments. Our response is shown below :
>
> > What would be the key component to benefit the tasks? And would it also benefit optical flow?
>
> The key technical design to benefit the task is the capturing of synergy between features, cost volumes, and their aggregations. While our approach to achieve this is indeed by incorporating self-attention and cross-attention mechanisms for effective aggregation, it's important to note that these are part of a suite of carefully considered technical designs. These designs are meticulously crafted to meet three key objectives: minimizing computational resources (achieved through Conv4D operations and maintaining a fixed resolution of the cost volume), maximizing performance (enhanced by residual connections among aggregated costs for training stability), and fostering synergy between feature aggregation and cost aggregation (via our innovative attention designs).
>
> However, we contend that these technical aspects, while significant, constitute a smaller part of our overall contribution. The true essence and value of our work lie in the comprehensive and thorough investigations we have conducted into the relationships and synergies between feature aggregation and cost aggregation, especially in the context of dense matching tasks. Our research goes beyond merely presenting a new technical design for state-of-the-art performance. Instead, it offers a deeper understanding of the underlying principles and interactions within these aggregation methodologies, as appreciated by the reviewer HY22. This in-depth exploration, we believe, provides a more substantial and lasting contribution to the field.
>
> This general approach, as the reviewer kindly suggested to show, is not constrained to geometric and semantic and it can also be evaluated  on other tasks, *e.g.,* optical flow, to further demonstrate the generalizability of our framework.  The results are shown below:
>
> | Methods  | KITTI-2015 | SINTEL (Final) |
> |----------|:----------:|-------:|
> | RAFT     |    5.32    |   2.69 |
> | PDC-Net  | 5.40       | 4.54   |
> | PDC-Net+ |    4.53    |      - |
> | GMFlow   | 7.77       | 3.44   |
> | Ours     | 4.39       | 4.16   |
>
> From the results, while ours outperforms others in KITTI-2015, it is observed that ours achieves lower AEPE than existing flow methods. However, it should be noted that when compared to PDC-Net and PDC-Net+, ours consistently performs better, which validates the effectiveness of our approach in optical flow task as well. We acknowledge that newer, more sophisticated technical designs may emerge in the future, but we wish to emphasize that while numerous architectural designs, e.g., convex upsampling, uncertainty module or GOCor-like correlation optimization scheme, could potentially be included and enhance the synergy between feature aggregation and cost aggregation, our framework works as a stepping stone that stands out for its simplicity and effectiveness.
>
>
> We highly appreciate the reviewer's constructive comment, and we will integrate the results into the final manuscript.
>
> > Limitations
>
> A pervasive challenge in dense matching methods is their requirement to extract all matches between views. This challenge often results in many dense matching networks facing difficulties in sparse or semi-sparse methods for geometry estimation. Our framework is not immune to this limitation. A specific area where our current architecture falls short is in its approach to handling confidence scores. While our cost volumes do encode a confidence score for each tentative match, a recent study like Lightglue has shown the benefits of explicitly estimating these confidence scores directly from the network, particularly for sparse matching tasks. This explicit estimation technique enhances performance in challenging scenarios, such as those involving occlusions or uncertain regions. Through empirical experiments, we've found that our model can struggle in these situations, largely due to the absence of a dedicated confidence estimation module.  To provide a complete understanding of our framework's capabilities and limitations, we will introduce a new section into our paper dedicated to discussing this limitation.

---

> ### Author Response · Authors · 2023-11-16
> **Response to the reviewer 96VU (2/3)**
>
> > Qualitative comparisons of Table 4.
>
>
> In response to the suggestion of including qualitative comparisons, we recognize the importance of such visualizations in complementing our quantitative results, as presented in Table 4. However, the qualitative results as the performance improves would primarily manifest as minor adjustments in the accuracy of warped keypoints or pixels. These changes, while significant in a quantitative sense, may not be visibly distinguishable. However, as an alternative, we present PCA visualization as in Fig.9 and Fig.10 in the supplementary material, where we provide more examples of how our method works. These visualizations are designed to provide an intuitive understanding of how our model achieves its performance gains. They offer a clear illustration of the internal workings and improvements brought about by our framework. For example, if we look at (I-IV), they semantically match with the change from raw features or cost volume to those that are aggregated, which are visualized in Fig.9 and Fig.10 in the supplementary material, while the rest (V-VI) do not provide much intuition but rather are similar to the already presented visualizations with being higher resolution (meaning more fine details).
>
> Nevertheless, to provide additional information for better understanding, we revised the supplementary material and included Fig.11, where we show PCA visualization comparisons of the each variant introduced in the component ablation table. The caption is colored red to draw attention to the change. As the comparison shows, while there exist apparent differences between (I) and (IV), we acknowledge that the differences are not visually apparent as we add the components from (II) to (IV). This is because the resolutions of features are defined in the coarsest level, which makes it little bit hard for us to interpret the visualization. Nevertheless, we believe that the differences present in the visualizations contributed to the improvements and that we provide more intuitive visualizations in Fig. 9 and Fig. 10.
>
> > It would be good to include the number of learnable parameters for Table 3 and Table 4.
>
> We appreciate the reviewer's suggestion. A small issue is that we intentionally adjusted all variants' number of learnable parameters to be similar for the experiments in Table 3 and 4 as mentioned in section 5.5, but we agree that it would be more informative to include the number of learnable parameters for each module, i.e., integrative self-attention and cross-attention, in our architecture.   The table below shows the comparison.
>
> |                                                    | Number of params |
> |----------------------------------------------------|:----------------:|
> | l = 1, integrative self-attention                  |      1612896     |
> | l = 1, cross-attention with matching distribution  | 308576           |
> | l = 2, integrative self-attention                  |      2511968     |
> | l = 2, cross-attention with matching distribution  | 683104           |
> | l = 3, integrative self-attention                  | 6513248          |
> | l = 3, cross-attention with matching distribution  | 2692960          |
> | l=1,2,3 cross-attention with matching distribution | 3684640          |
> | l=1,2,3 integrative self-attention                 | 10638112         |
> | Total attention params                             | 14.3 M           |
> | Total params                                       | 14.5 M           |
>
> Note that we also include similar ablation study in Table 2 of the supplementary material, which shows the comparison by varying depth of the transformer layers. We will include this in the supplementary material, as a separate table.

---

> ### Author Response · Authors · 2023-11-16
> **Response to the reviewer 96VU (3/3)**
>
> > Resolution of cost volumes at each level
>
> We apologize for any confusion caused by our initial explanations and appreciate the opportunity to clarify. In our architecture, as depicted in the upper branch of Figure 5, the cost volumes at each level are indeed fixed to the coarsest resolution. This specification applies specifically to the region labeled as "Cost Volume," where Conv4D operations are performed over the output cost volumes resulting from both self- and cross-attention mechanisms.
>
> Additionally, within the aggregation modules illustrated in Figure 4, the dimensions of the cost volumes are maintained to be consistent with those of the feature maps. This design choice ensures that the cost volumes and feature maps are compatible for effective integration within these modules.
>
> The rationale behind fixing the cost volume dimensions to the coarsest resolution in certain parts of the network, particularly for the residual connections, is driven by memory efficiency considerations. Since cost volumes inherently require more memory to store compared to feature maps, reducing their dimensions in specific regions of the network helps in managing the overall memory usage more effectively. This approach strikes a balance between maintaining the integrity and effectiveness of the cost volume information while optimizing the memory demands of the network.
>
> We hope this clarification addresses the concerns and provides a clearer understanding of our architectural choices and their underlying motivations. If there are further questions or aspects that require additional explanation, we are more than willing to provide the necessary information.

---

> > ### Comment · Area_Chair_pK98 · 2023-11-21
> > **Dear reviewer**
> >
> > Please try your best to engage within the discussion period.

---

> > ### Comment · Reviewer_96VU · 2023-11-23
> >
> > Thanks for the detailed responses! They resolved most of my concerns.
> >
> > One quick question: why do the visualized PCA features in Fig 10 and 11 in the supplemental seem to show distinctive sparse blobs, rather than dense smoothed visualization? If the method provides distinctive (or discriminative) features only for certain areas of the objects, can this method be suitable for the dense correspondence task? It would be also interesting to see a visualization of a dense matching field (or optical flow field) of the examples in Fig 10 and 11 in the supplemental.

---

> ### Author Response · Authors · 2023-11-23
> **Quick Reply**
>
> Thanks for the reply and we are glad to find that most of the concerns are resolved!
>
> Before answering your question, we first would like to ask if the reviewer wants to see PCA visualizations of feature maps (H x W x 3) for geometric matching or  the flow map (H x W x 2) itself.
>
> For the question, we can easily answer this, since it's because of the way visualize them by PCA, and the distinctive sparse blobs are the 3 principal components that represent the most distinctive ones. Other components represent the other regions well, but one thing is that they just are not very intuitive. We have a few PCA visualizations for geometric matching datasets ready, and if this is what the reviewer wants to see, we can quickly revise the supplementary material. Note that these were also not very intuitive, visualizing some horizontal or vertical lines (we hypothesize that they indicate flows).
>
> Please let us know and we will get back to you as soon as possible!

---

> > ### Comment · Reviewer_96VU · 2023-11-30
> >
> > @Authors: I was wondering if it's possible to visualize the flow map (H x W x 2) to see the method is good at dense correspondence (as the title claims) in Fig 10 and 11 in the supplemental.
> > The warping results in Fig. 3 in the supplementary looks great, but it uses "quasi-dense flow by COTR+interp" so it doesn't really tell how good the dense correspondence quality is.

---

### Author Response · Authors · 2023-11-16
**General Response**

We express our sincere gratitude to the reviewers for their constructive feedback and suggestions. We are particularly appreciative of the acknowledgment of our results' robustness (96VU, 4Eyk, gxEA), the visualizations provided were found to be effective in elucidating the workings of our method and in validating its efficacy (4Eyk, HY22), as well as the recognition of the extensive analysis and experimentation that underpins our study (96VU, 4EyK, HY22). We are also encouraged by the reviewers' interest in the motivation behind our research (4Eyk, HY22), and we are pleased to note the acknowledgment that our work represents the first to explore the relationship between two popular techniques in this field (HY22).

To re-emphasize our contribution: While feature aggregation and cost aggregation are widely recognized techniques in computer vision, their interrelationship, particularly in the context of dense matching, has not been extensively explored. Our work pioneers this area by thoroughly investigating the intricate dynamics between these two methodologies and their synergistic potential. To capture such synergy, we have developed a novel yet generalizable framework that adeptly integrates feature and cost volume aggregation to address the challenges inherent in dense matching. To validate our framework's efficacy and versatility, we conducted comprehensive evaluations across two distinct domains: semantic matching and geometric matching, where we clearly achieve competitive performance. The results from these evaluations not only underscore the effectiveness of our approach but also highlight its applicability across different matching scenarios, thereby substantiating our contribution to the field.

---

### Author Response · Authors · 2023-11-21
**Gentle reminder**

Dear reviewers,

Thank you again for volunteering time to review our manuscript and helping to improve it. Since the discussion period nears its end, we would like to kindly invite the reviewers to check our responses.

- **96VU**: We are highly appreciated for the reviewer’s constructive revises, which include suggestion to include additional results on optical flow, discussion of limitations and more qualitative results for Table 4. In the rebuttal, we have included the new results on optical flow, which are proven to be effective and demonstrates the generalization power of the unified aggregation approach we propose, and we have discussed a limitation. We also revised our supplementary material to include more qualitative results that will help the readers’ understanding. We really hope that the reviewer will reconsider reassessing the importance and the value of the thorough investigations we conducted.

- **gxEA**: We thank the reviewer for the thorough reviews. Addressing all the comments that concerns about our experiments, we have included new results for all the tasks the reviewer suggested. We have also included some discussions regarding the state-of-the-art performance, which is an important aspect. We highlight the true essence and value of our work lie in the comprehensive and thorough investigations we have conducted into the relationships and synergies between feature aggregation and cost aggregation, as appreciated by the reviewer **HY22**. We hope that the reviewer reevaluate the significance and merit of the comprehensive investigations we have undertaken.

- **HY22**: We appreciate the reviewer’s detailed comments regarding the presentation of our work. In this rebuttal, we have re-explained some of the ambiguous explanations or terminologies that might have caused confusions, which we will reflect the reviewer’s comments in our final manuscript. Regarding one comment about experimental comparison, we have included two tables to compare with more recent work and GMFlow in optical flow benchmark, where ours demonstrate its effectiveness. Recognizing the considerable workloads that reviewers typically manage, we earnestly hope that the reviewer will contribute to the discussion.

---

### Author Response · Authors · 2023-11-23
**Final Response**

Dear Reviewers,

We highly appreciate your efforts and time in reviewing our manuscript.
As the author-reviewer discussion is about to end, we would like to make a final response that may help the further discussion.

We wish to highlight that the main contribution of our work lies in the **thorough investigation of feature and cost aggregation** and the extensive experiments we conducted to demonstrate the effectiveness and the generalization power of the proposed approach to unify both aggregations.

We wish to clarify that our method is specifically tailored for challenging dense correspondence tasks, including intra-class variations, repetitive patterns or large displacements, thus evaluating on 5 datasets. However, in response to the reviewers' suggestions,  although they do not align with our primary objective, as they either depict scenes with small displacements or are often used for sparse matching tasks like pose estimation, we have shown additional experimental results on MegaDepth, YFCC100M, ScanNet, KITTI and SINTEL, where our work consistently shows competitive performance. While for some of them, ours does not attain SOTA, we have also included a discussion about this.

We believe that this submission does not lack analysis or experiments, and it has sufficiently demonstrated its effectiveness with extensive experiments. In terms of performance, it is also outperforming, even compared to the concurrent submission in ICLR [A], where their experiments are acknowledged by the reviewers with high scores, and **they tackle the same task and evaluated on the datasets that overlap with ours**. We sincerely hope that the reviewers consider these for the subsequent discussion period.

[A] @inproceedings{
anonymous2023diffusion,
title={Diffusion Model for Dense Matching},
author={Anonymous},
booktitle={Submitted to The Twelfth International Conference on Learning Representations},
year={2023},
url={https://openreview.net/forum?id=Zsfiqpft6K },
note={under review}
}

---

### Meta-Review · Area_Chair_pK98 · 2023-12-06

**Metareview:**

The paper analyzes feature aggregation, where similar features both within an image and across the two images are aligned, and cost aggregation, where the flow predictions between neighboring pixels are compared and then aligned, e.g. smoothing the final flow field. The paper finds that these two types of aggregation interact in a particular way; specifically, more robust descriptors can construct a less noisy cost volume but typically due to network structure the cost volume does not affect the feature set. To fix this, the paper recommends a "integrative" aggregation where the feature extraction and the cost computation are done in a way that makes them aware of each other. In addition to this, the paper also adds a coarse-to-fine formulation. The method outperforms several baselines on semantic correspondences (SPair-71k, PF-Pascal, PF-Willow) and geometric matching (HPatches, ETH3D). An additional ablation study also show that each component contributes to the performance gain. The intuition and proposed solution seem solid and the majority of the results seem to suggest that the method work. In general, the experimental evidence seems relatively thorough.

**Justification For Why Not Higher Score:**

While the idea is solid, it shares similarity with a large number of prior works. The results seem good but not excellent.

**Justification For Why Not Lower Score:**

The paper seems to offer a clean idea with enough experimentation to back up the relatively simple claim.

---

### Decision · Program_Chairs · 2024-01-16

Accept (poster)